# The histone deacetylase complex MiDAC regulates a neurodevelopmental gene expression program to control neurite outgrowth

Baisakhi Mondal[1], Hongjian Jin[2], Satish Kallappagoudar[1], Yurii Sedkov[1], Tanner Martinez[1], Monica F Sentmanat[3], Greg J Poet[4†], Chunliang Li[4], Yiping Fan[2], Shondra M Pruett-Miller[1], Hans-Martin Herz[1]*

[1]Department of Cell & Molecular Biology, St. Jude Children's Research Hospital, Memphis, United States; [2]Department of Computational Biology, St. Jude Children's Research Hospital, Memphis, United States; [3]Genome Engineering & iPS Center, Department of Genetics, Washington University, St. Louis, United States; [4]Department of Tumor Cell Biology, St. Jude Children's Research Hospital, Memphis, United States

*For correspondence:
hans-martin.herz@stjude.org

Present address: †The Institute of Molecular, Cell and Systems Biology, University of Glasgow, Glasgow, United Kingdom

Competing interests: The authors declare that no competing interests exist.

**Abstract** The mitotic deacetylase complex (MiDAC) is a recently identified histone deacetylase (HDAC) complex. While other HDAC complexes have been implicated in neurogenesis, the physiological role of MiDAC remains unknown. Here, we show that MiDAC constitutes an important regulator of neural differentiation. We demonstrate that MiDAC functions as a modulator of a neurodevelopmental gene expression program and binds to important regulators of neurite outgrowth. MiDAC upregulates gene expression of pro-neural genes such as those encoding the secreted ligands SLIT3 and NETRIN1 (NTN1) by a mechanism suggestive of H4K20ac removal on promoters and enhancers. Conversely, MiDAC inhibits gene expression by reducing H3K27ac on promoter-proximal and -distal elements of negative regulators of neurogenesis. Furthermore, loss of MiDAC results in neurite outgrowth defects that can be rescued by supplementation with SLIT3 and/or NTN1. These findings indicate a crucial role for MiDAC in regulating the ligands of the SLIT3 and NTN1 signaling axes to ensure the proper integrity of neurite development.

## Introduction

Epigenetic regulators often constitute chromatin-modifying enzymes which catalyze the addition and removal of posttranslational modifications on histones and are involved in the control of gene expression by modulating chromatin structure and function. Among them, the families of histone acetyltransferases (HATs) and histone deacetylases (HDACs) acetylate and deacetylate lysine residues on histones and other proteins, respectively (*Shahbazian and Grunstein, 2007*). Based on initial studies, a model emerged in which nuclear HDACs were thought to be recruited by transcription factors to facilitate transcriptional repression by creating a more condensed chromatin landscape at their target genes (*Kadosh and Struhl, 1998*; *Rundlett et al., 1998*; *Yang et al., 1996*; *Yang et al., 1997*). However, more recently it was shown that HDACs tend to localize to transcriptionally active loci including promoters, gene bodies and enhancers (*Wang et al., 2009*). Additionally, gene expression profiling studies also demonstrated that knockout of individual HDACs resulted not only in upregulation but also downregulation of a significant number of genes suggesting both activating

and repressive roles for HDACs in regulating gene transcription (*Bernstein et al., 2000*; *Harrison et al., 2011*; *Yamaguchi et al., 2010*; *Zupkovitz et al., 2006*).

HDAC1 and HDAC2 belong to the family of class I HDACs, are highly similar (83% identity) and are involved in the control of gene expression by modulating chromatin structure and function. Their vital and often redundant roles in neurogenesis are well established (*Chen et al., 2011*; *Gräff et al., 2012*; *Guan et al., 2009*; *Jacob et al., 2011*; *Kim et al., 2010*; *Montgomery et al., 2009*; *Ye et al., 2009*). They form the catalytic core of the SIN3, NuRD, CoREST and mitotic deacetylase (MiDAC) complexes (*Kelly and Cowley, 2013*; *Millard et al., 2017*). HDAC1/2 activity and targeting to specific gene loci strongly depends on their incorporation into these complexes (*Kelly and Cowley, 2013*; *Millard et al., 2017*). At least one of the scaffolding proteins within the NuRD, CoREST and MiDAC complexes contains an ELM2-SANT domain which is instrumental in recruiting and activating HDAC1/2 (*Millard et al., 2017*). While the molecular function of the SIN3, NuRD and CoREST complexes have been studied in greater detail little is known about the molecular function of MiDAC (*Kelly and Cowley, 2013*; *Millard et al., 2017*).

MiDAC is conserved from nematodes to humans (*Bantscheff et al., 2011*; *Hao et al., 2011*). In humans, MiDAC is composed of HDAC1/2, the scaffolding protein DNTTIP1, and the ELM2-SANT domain containing scaffolding protein ELMSAN1 (also known as MIDEAS) and/or the closely ELM-SAN1-related proteins TRERF1 and ZNF541 (ZFP541 or SHIP in mice) (*Figure 1A*; *Banks et al., 2018*; *Bantscheff et al., 2011*; *Choi et al., 2008*; *Hao et al., 2011*; *Joshi et al., 2013*). MiDAC constitutes a stoichiometric tetrameric complex that requires the N-terminal dimerization domain of DNTTIP1 for assembly. The C-terminus of DNTTIP1 mediates MiDAC recruitment to nucleosomes in vitro (*Itoh et al., 2015*). MiDAC specifically associates with cyclin A2 (CCNA2) and cyclin dependent kinase 2 (CDK2) and interacts more prominently with certain HDAC inhibitors in mitotically arrested versus non-synchronized proliferating cells, suggesting a role in cell cycle regulation (*Bantscheff et al., 2011*; *Hein et al., 2015*; *Huttlin et al., 2015*; *Pagliuca et al., 2011*). Before MiDAC was described as a complex its individual subunits DNTTIP1 and TRERF1 were reported to function predominantly as transcriptional activators for select genes of the steroidogenesis and ossification pathways, respectively (*Gizard et al., 2001*; *Gizard et al., 2002*; *Gizard et al., 2005*; *Gizard et al., 2006*; *Gizard et al., 2004*; *Koiwai et al., 2015*; *Kubota et al., 2013*). In agreement with these findings, ELMSAN1 was found to be associated with chromatin enriched for the histone mark H3K27ac suggesting a role in active transcription (*Ji et al., 2015*). While SIN3, NuRD, and CoR-EST have been functionally characterized as regulators of neurogenesis, the physiological role of MiDAC remains unexplored (*Andrés et al., 1999*; *Knock et al., 2015*; *Nitarska et al., 2016*; *Wang et al., 2016*).

To decipher the physiological function of MiDAC, we employed mESCs as an experimental model system. We found that MiDAC is recruited to promoters and enhancers genome-wide, directly regulates a set of neurodevelopmental genes, and can act as both an activator and repressor of different gene sets to modulate gene expression by negatively regulating the repressive histone mark H4K20ac or the active histone mark H3K27ac, respectively. Specifically, MiDAC binds to promoters and enhancers of axon guidance ligands of the SLIT and NETRIN families and is required for their activation. Genetic deletion of MiDAC components results in inactivation of SLIT3 and NTN1 signaling during neural differentiation and severely impairs neurite outgrowth and network formation. These findings reveal a novel function for MiDAC in regulating neural gene expression programs to ensure proper neuronal maturation and/or neurite outgrowth during neurogenesis.

## Results

### DNTTIP1 interacts with ELMSAN1 and the histone deacetylase HDAC1 to form MiDAC in mESCs

We began by investigating the function of DNTTIP1 and ELMSAN1, two scaffolding components of MiDAC, in mESCs (*Figure 1A*). Firstly, we generated and validated CRISPR/Cas9-mediated knock outs (KOs) of *Dnttip1* and *Elmsan1* in mESCs (*Figure 1B*; *Figure 1—figure supplement 1A-F*; *Supplementary file 1*). Indels in the identified *Dnttip1* KO and *Elmsan1* KO clones resulted in the introduction of a premature stop codon downstream, producing KO clones with a complete loss of DNTTIP1 or ELMSAN1 protein compared to wild-type (WT) mESCs, respectively (*Figure 1B*;

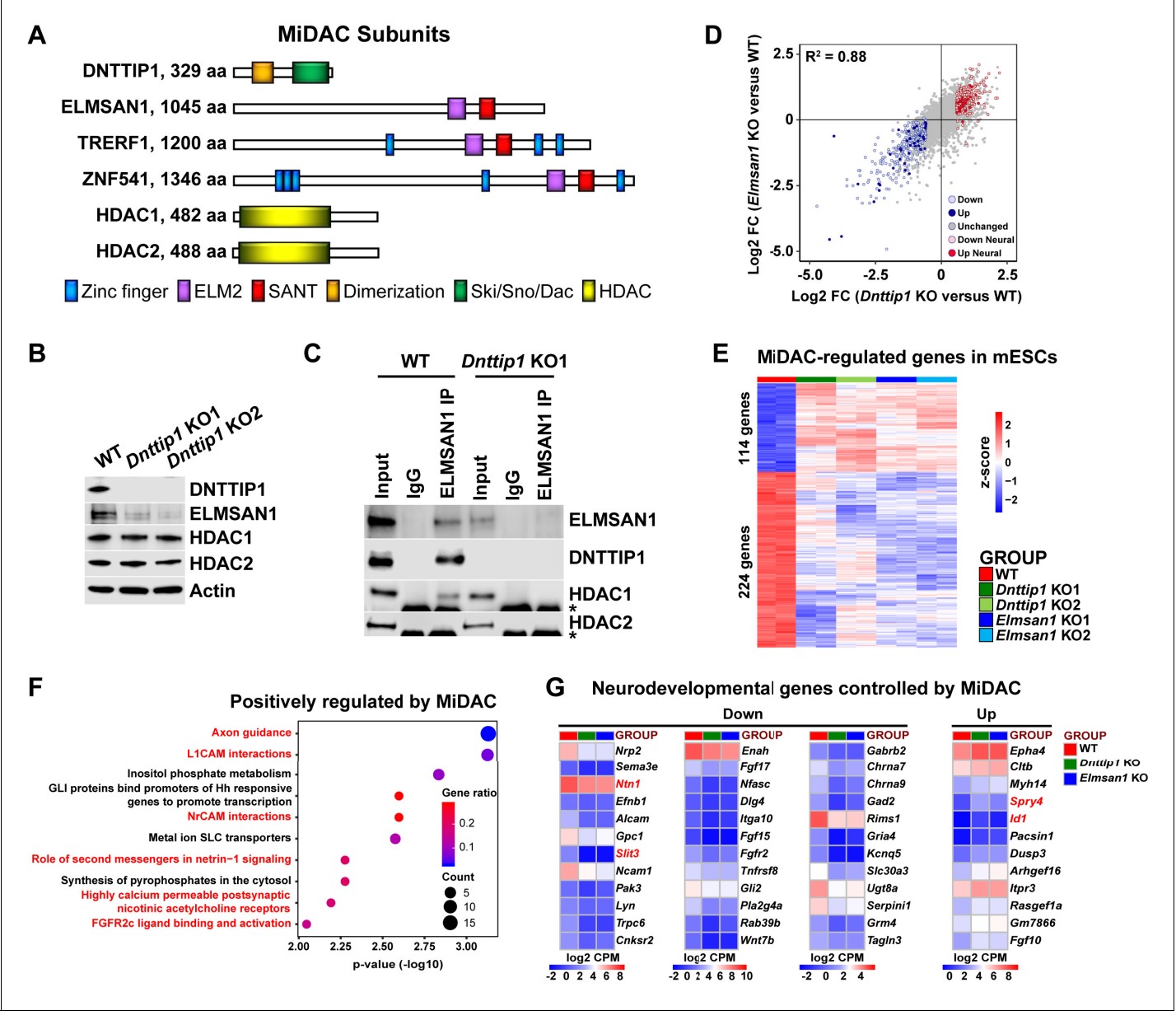

**Figure 1.** MiDAC controls a neurodevelopmental gene expression program. (**A**) Subunits with domains of the human histone deacetylase complex MiDAC. (**B**) WB for the indicated MiDAC components from total cell lysates of WT, *Dnttip1* KO1 and *Dnttip1* KO2 mESCs. Actin is the loading control. (**C**) IPs were carried out with IgG and ELMSAN1 antibodies from nuclear extracts of WT and *Dnttip1* KO1 mESCs followed by WB for the indicated MiDAC components. The asterisk marks the IgG heavy chain. (**D**) Scatter plot comparing all DEGs in *Dnttip1* KO (KO1 and KO2) versus WT mESCs from *Figure 1—figure supplement 2A* (x-axis) with DEGs in *Elmsan1* KO (KO1 and KO2) versus WT mESCs from *Figure 1—figure supplement 2B* (y-axis). Both axes depict normalized gene expression ($\log_2$ FC of CPM). (**E**) RNA-seq heatmap depicting DNTTIP1 and ELMSAN1 co-regulated (MiDAC-regulated) genes in mESCs (fold change (FC) >1.5 or <−1.5, p<0.01). The color scale depicts normalized gene expression ($\log_2$ CPM). (**F**) Reactome analysis showing the most highly enriched gene categories of genes that are positively regulated by MiDAC (both down in *Dnttip1* KO (KO1 and KO2) and *Elmsan1* KO (KO1 and KO2) versus WT mESCs, FC <−1.5, p<0.01). Pathways associated with neural differentiation and function are highlighted in red. (**G**) RNA-seq heatmaps depicting down- and upregulated genes from a gene set of neurodevelopmental genes that is mutually regulated by DNTTIP1 and ELMSAN1 (FC <−1.5 or >1.5, p<0.05). The color scale depicts the z-score of normalized gene expression ($\log_2$ CPM). (**D**) DEGs, (**E**) MiDAC-regulated genes, (**F**) Reactome gene categories and (**G**) differentially expressed neurodevelopmental genes were determined based on two biological replicates each from WT mESCs and two *Dnttip1* KO and two *Elmsan1* KO clones, respectively.

The online version of this article includes the following figure supplement(s) for figure 1:

**Figure supplement 1.** Characterization of *Dnttip1* KO and *Elmsan1* KO mESCs.

**Figure supplement 2.** MiDAC controls a neurodevelopmental gene expression program.

*Figure 1 continued on next page*

*Figure 1 continued*

**Figure supplement 3.** MiDAC is dispensable for self-renewal, pluripotency and cell cycle distribution of mESCs.

*Figure 1—figure supplement 1A,B,F*). To determine whether DNTTIP1 and ELMSAN1 function within MiDAC in mESCs, we examined the mRNA and protein levels of other MiDAC components in *Dnttip1* KO and *Elmsan1* KO mESCs (*Bantscheff et al., 2011*; *Choi et al., 2008*; *Hao et al., 2011*). *Elmsan1* and *Hdac1* mRNA levels were unchanged, *Hdac2* mRNA levels slightly reduced and *Dnttip1* mRNA levels more strongly reduced in *Dnttip1* KO versus WT mESCs (*Figure 1—figure supplement 1C and E*). However, ELMSAN1 protein levels were severely reduced with no significant effects on HDAC1 or HDAC2 protein levels in *Dnttip1* KO versus WT mESCs (*Figure 1B*). Compared to WT mESCs, *Elmsan1* KO mESCs showed no significant differences in *Elmsan1*, *Dnttip1*, *Hdac1*, or *Hdac2* mRNA levels (*Figure 1—figure supplement 1D and E*); at the protein level, HDAC1 and HDAC2 appeared unaltered, but DNTTIP1 levels were significantly lower (*Figure 1—figure supplement 1F*). These results suggest that in mESCs MiDAC becomes destabilized upon loss of its core subunits, DNTTIP1 or ELMSAN1. Immunoprecipitation (IP) of ELMSAN1 in WT and *Dnttip1* KO mESCs confirmed the existence of an HDAC1-containing MiDAC complex (*Figure 1C*). Furthermore, consistent with the ELMSAN1 IP in WT and *Dnttip1* KO mESCs, IP of DNTTIP1 in WT and *Elmsan1* KO mESCs corroborated the presence of an HDAC1-containing MiDAC complex in mESCs (*Figure 1—figure supplement 1G*). However, we did not observe an interaction between ELMSAN1 and HDAC2 and DNTTIP1 and HDAC2 in mESCs, suggesting that only DNTTIP1, ELMSAN1 and HDAC1 interact to form MiDAC in mESCs (*Figure 1C*; *Figure 1—figure supplement 1G*).

## MiDAC controls a neurodevelopmental gene expression program

To determine transcriptional targets of MiDAC, we performed RNA sequencing (RNA-seq) in WT, *Dnttip1* KO and *Elmsan1* KO mESCs. We identified 493 downregulated and 368 upregulated genes in *Dnttip1* KO versus WT mESCs (*Figure 1—figure supplement 2A*; *Supplementary file 2*). Differential gene expression analysis via RNA-seq revealed 467 downregulated and 306 upregulated genes in *Elmsan1* KO versus WT mESCs (*Figure 1—figure supplement 2B*; *Supplementary file 2*). The gene expression patterns between *Dnttip1* KO and *Elmsan1* KO mESCs were highly correlated ($R^2$ = 0.88) with many differentially expressed genes (DEGs) shared between *Dnttip1* KO and *Elmsan1* KO mESCs (*Figure 1D*; *Supplementary file 2*). Of the DEGs in *Dnttip1* KO and *Elmsan1* KO mESCs a common set of 114 upregulated and 224 downregulated genes was identified (*Figure 1E*; *Figure 1—figure supplement 2C*). Pathway analysis of this MiDAC-regulated gene set revealed an enrichment for axon guidance signaling and neural development related genes among the downregulated category (*Figure 1F*). Similar results were also obtained when DEGs in only *Dnttip1* KO or only *Elmsan1* KO mESCs were analyzed independently of each other (*Figure 1—figure supplement 2D-G*). Downregulated gene classes were generally associated with promoting neural differentiation (e.g., nervous system, axon guidance and GABAergic neurogenesis), whereas upregulated genes showed enrichment for pathways associated with repression of neuronal development (e.g., antagonists of nerve growth factor (NGF) signaling) (*Figure 1—figure supplement 2D-G*; *Supplementary file 2*). Importantly, we identified a set of neurodevelopmental genes that is associated with neurogenesis, axon guidance, and neurotransmitter receptor signaling and co-regulated by both ELMSAN1 and DNTTIP1 (*Figure 1D and G*). However, loss of MiDAC function did not affect self-renewal or pluripotency as evaluated by assessing the mRNA levels of the pluripotency factors *Sox2*, *Pou5f1*, and *Nanog* (*Figure 1—figure supplement 3A and B*), nor did it alter the percentage of alkaline phosphatase (AP)-positive colonies (*Figure 1—figure supplement 3C*). Furthermore, DNTTIP1 or ELMSAN1 loss in mESCs did not alter the proliferation rate or cell cycle profile (*Figure 1—figure supplement 3D and E*), in contrast to the reported growth-promoting function of DNTTIP1 in oral and non-small cell lung cancer (*Sawai et al., 2018*; *Zhang et al., 2018*). Gene expression of *Ccna2* and the cell cycle inhibitors *Cdkn1a* (*p21*) and *Cdkn1b* (*p27*) whose protein products were reported to interact with and/or be regulated by DNTTIP1 were also unaffected in *Dnttip1* KO and *Elmsan1* KO mESCs (*Figure 1—figure supplement 3F*; *Hein et al., 2015*; *Huttlin et al., 2015*; *Pagliuca et al., 2011*; *Sawai et al., 2018*). Together, this suggests that loss of MiDAC function does not impair self-renewal and pluripotency in mESCs and that MiDAC's role in

controlling proliferation and the cell cycle is largely cell type or context-dependent. Rather, our findings indicate that DNTTIP1 and ELMSAN1 regulate a highly similar set of target genes in mESCs and that MiDAC controls a neurodevelopmental gene expression program, including genes that are important regulators of neurite outgrowth and morphogenesis.

## MiDAC regulates neurite outgrowth

To investigate the function of MiDAC in neuronal development, we differentiated mESCs into neuro-ectoderm (NE) (*Figure 2A*; *Li et al., 2009*). We evaluated WT, *Dnttip1* KO and *Elmsan1* KO NE after 8 days of differentiation, a time point that corresponds to the initiation of differentiation into neural progenitor cells (NPCs), which is accompanied by increased expression of *nestin* (*Nes*) and *Pax6* as well as increased protein levels of the early NE differentiation markers, PAX6 and MASH1. No significant differences in mRNA and protein levels of the above-mentioned NPC markers were detected between WT and *Dnttip1* KO or *Elmsan1* KO cells at day 0 or day 8 (*Figure 2—figure supplement 1A-D*). Moreover, PAX6-positive cells from WT, *Dnttip1* KO and *Elmsan1* KO NE at day 8 displayed a similar cell cycle profile, indicating no differences in cell cycle distribution in NPCs (*Figure 2—figure supplement 1E*). These results demonstrate that loss of MiDAC function in mESCs does not affect the initiation of differentiation into the NE lineage. Upon extended differentiation through day 12, many WT cells underwent extensive morphological changes and developed into mature neurons with long neurites while the neurite length of *Dnttip1* KO and *Elmsan1* KO neurons was substantially shorter (*Figure 2B*). Consistent with these observations, *Dnttip1* KO and *Elmsan1* KO neurons showed significantly reduced *Tubb3* and *Map2* mRNA levels (immature and mature neuron marker genes, respectively) compared to WT neurons, suggesting that loss of MiDAC function affects neuronal maturation and/or differentiation (*Figure 2C and D*). *Dnttip1* KO and *Elmsan1* KO neurons also showed a significant decrease in the average neurite length per neuron as well as the total number of neurites per neuron compared to WT neurons as assessed by MAP2 staining at day 12 of differentiation (*Figure 2B,E,F*). However, the percentage of neurons within the total differentiated cell population remained unchanged between WT and *Dnttip1* KO or *Elmsan1* KO NE (*Figure 2G*; *Figure 2—figure supplement 1F*). We next performed qRT-PCR at day 12 for a subset of NE genes involved in neurite outgrowth and morphogenesis. Similar to our findings in mESCs, *Dnttip1* KO and *Elmsan1* KO NE showed decreased expression of several positive regulators of neuronal development and axon guidance (e.g., *Slit3*, *Ntn1*, *Ncam1*) as well as increased expression of repressors of neuronal development (e.g., *Spry4*, *Id1*, *Pacsin1*) compared to WT neurons (*Figure 2H*). Overall, our data suggest that MiDAC is required to transcriptionally control a gene set of regulators that are important for neuronal maturation or neurite outgrowth but that loss of MiDAC function does not affect initiation of neuronal differentiation per se.

## MiDAC binds to and modulates the expression of genes that regulate neural differentiation and neurite outgrowth

To better understand how MiDAC regulates gene expression in mESCs, we performed ChIP sequencing (ChIP-seq) in WT, *Dnttip1* KO and *Elmsan1* KO mESCs to map the genome-wide occupancy of DNTTIP1. This analysis identified 61,505 DNTTIP1-specific peaks, of which 22% localized within +/- 1 kb of a transcription start site (TSS) (*Figure 3A*; *Supplementary file 3*). The remaining 78% of DNTTIP1 binding sites were located upstream of (24%), downstream of (12%), or within (43%) a gene. Of the 47,657 non-TSS DNTTIP1 peaks, a majority (84%) was found at a distance of >5 kb from the TSS (*Figure 3A*; *Supplementary file 3*). Interestingly, a significant number of DNTTIP1 binding sites overlapped with the active histone marks H3K4me1 and/or H3K27ac and TSS-associated DNTTIP1 peaks were strongly enriched for the active histone mark H3K4me3 along with H3K27ac and H3K4me1 (*Figure 3B*; *Figure 3—figure supplement 1A*). Furthermore, low DNTTIP1 binding at TSS correlated with increased enrichment of the repressive histone mark H3K27me3 (*Figure 3—figure supplement 1A*). Non-TSS DNTTIP1 binding sites were enriched for the enhancer marks H3K4me1 and H3K27ac, hinting at a potential role for DNTTIP1 in enhancer-mediated processes (*Figure 3—figure supplement 1A*). We next correlated our DNTTIP1 ChIP-seq data with gene expression data from *Dnttip1* KO mESCs. This analysis revealed that out of 493 downregulated genes in *Dnttip1* KO mESCs, 358 genes (73%) were bound by DNTTIP1, while out of 368 upregulated genes in *Dnttip1* KO mESCs, 314 genes (85%) were occupied by DNTTIP1 at either

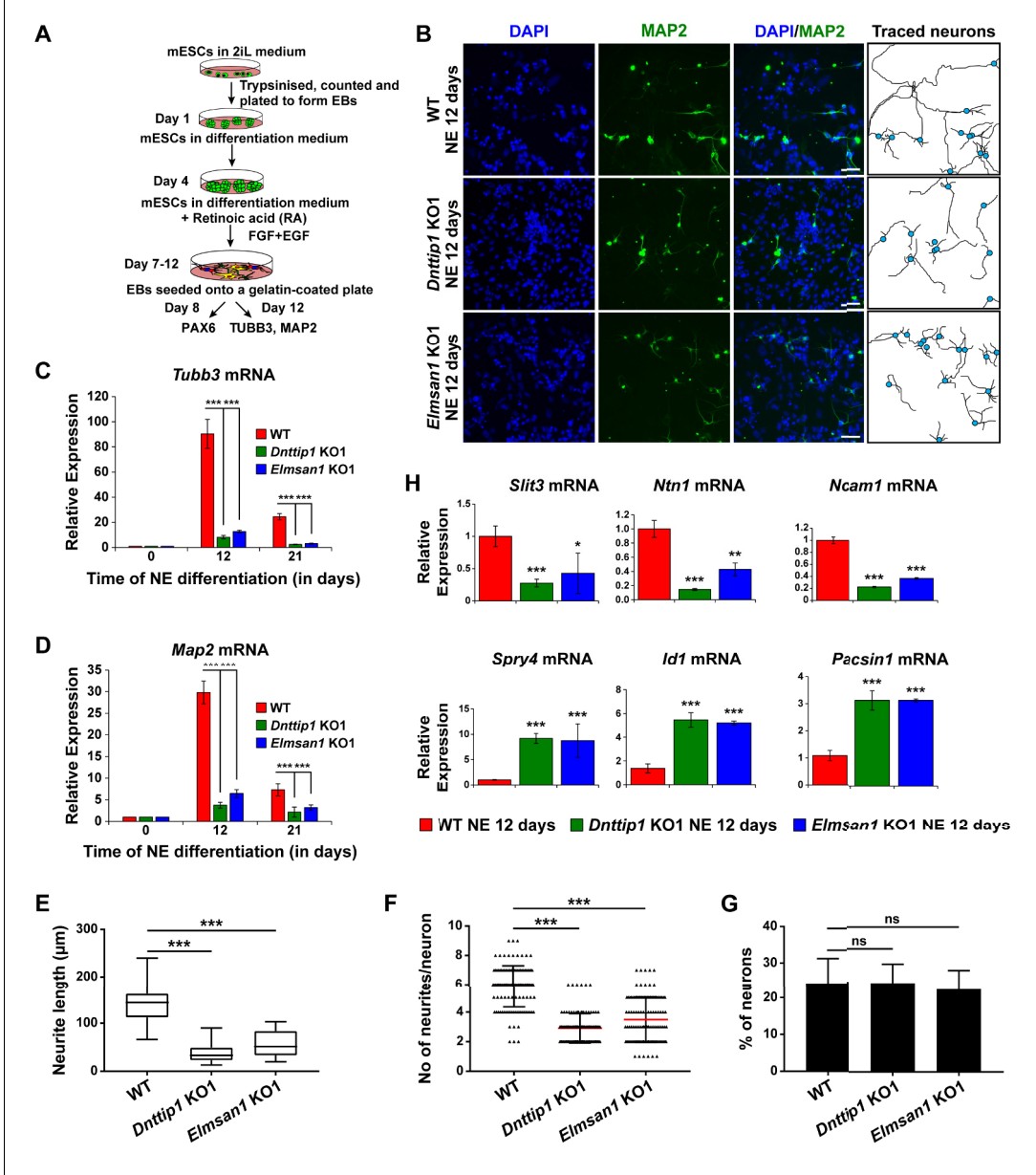

**Figure 2.** MiDAC regulates neurite outgrowth. (**A**) Schematic outline of neuro-ectoderm (NE) differentiation protocol. (**B**) MAP2 immunofluorescence (IF) staining of WT, *Dnttip1* KO1 and *Elmsan1* KO1 NE after 12 days of differentiation. Nuclei were stained with DAPI. For analysis the neuronal cell body (blue) and its neurites were manually traced with ImageJ software. The white scale bar represents 20 μm. (**C, D**) qRT-PCR for (**C**) *Tubb3* and (**D**) *Map2* mRNA in WT, *Dnttip1* KO1 and *Elmsan1* KO1 mESCs (day 0) and NE after 12 and 21 days of differentiation. Expression was normalized to *Gapdh*. (**E–G**) Quantification of (**E**) neurite length, (**F**) the total number of neurites per neuron and (**G**) the percentage of neurons within the total cell population from traced neurites in WT, *Dnttip1* KO1 and *Elmsan1* KO1 neurons after 12 days of differentiation as determined by MAP2 IF. (**E, F**) The neurites of 200 neurons within the total cell population were assessed per genotype. (**H**) Gene expression levels of the indicated neurodevelopmental genes in WT, *Dnttip1* KO1 and *Elmsan1* KO1 NE after 12 days of differentiation as analyzed by qRT-PCR. Expression was normalized to *Gapdh*. Unpaired t-test was performed throughout where ***, p≤0.001; **, p≤0.01; *, p≤0.05; and ns, p>0.05 is not significant.

The online version of this article includes the following figure supplement(s) for figure 2:

**Figure supplement 1.** Loss of MiDAC function does not alter the differentiation and cell cycle distribution of neural progenitor cells.

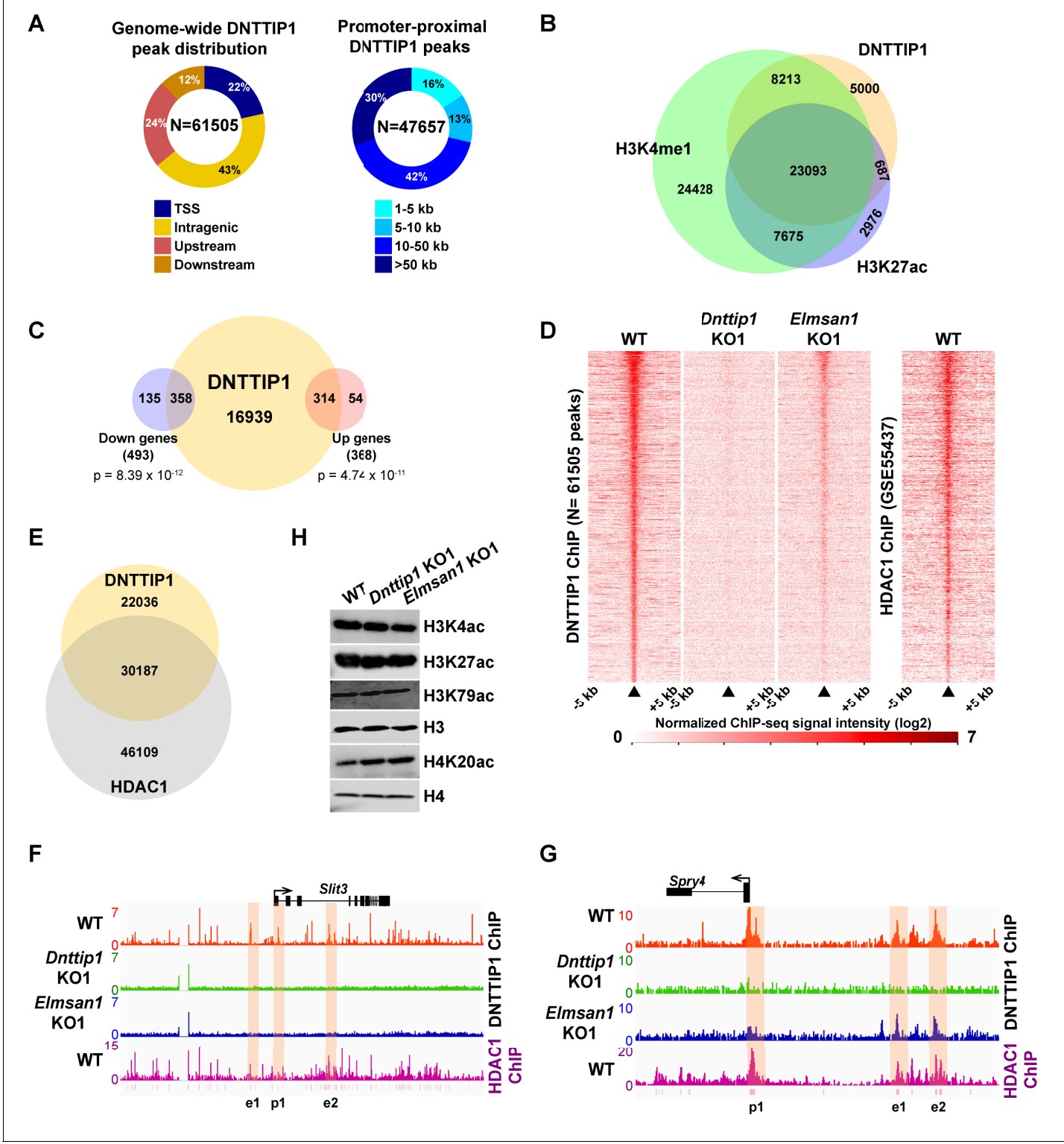

**Figure 3.** MiDAC binds to and modulates the expression of genes that regulate neural differentiation and neurite outgrowth. (A) Pie charts displaying the genome-wide distribution (left) and promoter distal distribution (right) of DNTTIP1 in WT mESCs. DNTTIP1 peaks within 1 kb of the transcription start site (TSS) were assigned to TSS. (B) Venn Diagram depicting the overlap between DNTTIP1, H3K4me1 and H3K27ac peaks in WT mESCs. (C) Venn diagram showing the number of downregulated and upregulated genes in *Dnttip1* KO (KO1 and KO2) versus WT mESCs that are bound by DNTTIP1 in WT mESCs. (D) Heatmaps displaying the genome-wide distribution of all DNTTIP1 binding sites in WT, *Dnttip1* KO1 and *Elmsan1* KO1 mESCs sorted by enrichment in descending order in WT mESCs and compared to HDAC1 occupancy in WT mESCs. The color scale depicts the normalized ChIP-seq

*Figure 3 continued on next page*

Figure 3 continued

signal intensity (log$_2$ CPM per 20 bp bin). (E) Venn diagram showing the co-occupancy between DNTTIP1 and HDAC1 peaks in WT mESCs. (A–E) DNTTIP1, H3K4me1 and H3K27ac ChIP-seq peaks were determined based on the average of two replicates each from WT mESCs and where applicable based on the average of two replicates each from *Dnttip1* KO1 and *Elmsan1* KO1 mESCs, respectively. HDAC1 ChIP-seq data were obtained from GSE55437. Downregulated and upregulated genes were determined based on two biological replicates each from WT mESCs and two *Dnttip1* KO (KO1 and KO2) clones. (F, G) ChIP-seq profiles of the (F) *Slit3* and (G) *Spry4* loci for DNTTIP1 in WT, *Dnttip1* KO1 and *Elmsan1* KO1 mESCs and for HDAC1 in WT mESCs. Promoter and putative enhancer regions used for manual ChIP experiments in *Figure 4* are highlighted by orange boxes. The track files depict the average of two ChIP-seq replicates each from WT mESCs and the average of two replicates each from the *Dnttip1* KO1 and *Elmsan1* KO1 clone, respectively. (H) WB for the specified histone acetylation marks from total cell lysates of WT, *Dnttip1* KO1 and *Elmsan1* KO1 mESCs. H3 and H4 are loading controls.

The online version of this article includes the following figure supplement(s) for figure 3:

**Figure supplement 1.** MiDAC binds to and modulates the expression of genes that regulate neural differentiation and neurite outgrowth.

their promoters or other gene regulatory regions such as enhancers (*Figure 3C*; *Figure 3—figure supplement 1B and C*; *Supplementary file 4*). Importantly, an even higher proportion of differentially expressed neurodevelopmental genes was also bound by DNTTIP1 (36 out of 39 (92%) in the downregulated and 22 out of 23 (96%) in the upregulated category) (*Supplementary file 4*). This overlap is highly significant and indicates that the majority of DEGs in *Dnttip1* KO mESCs constitute direct targets of DNTTIP1, including important regulators of neuronal differentiation and neurite outgrowth. Moreover, motif analysis of DNTTIP1 peaks revealed that DNTTIP1 binding sites associated with down- or upregulated genes in *Dnttip1* KO mESCs are selectively enriched for transcription factor (TF) binding motifs, including some that have been implicated in neurogenesis (*Figure 3—figure supplement 1D*). These TFs include RBFOX2 (associated with genes upregulated in *Dnttip1* KO mESCs) and ELK1, a member of the ETS TF family (associated with genes downregulated in *Dnttip1* KO mESCs) (*Figure 3—figure supplement 1D*; *Besnard et al., 2011*; *Gehman et al., 2012*). We further assessed the relationship between DNTTIP1 and ELMSAN1 by determining the genome-wide occupancy pattern of DNTTIP1 in *Elmsan1* KO mESCs. While no DNTTIP1 binding to chromatin could be detected in *Dnttip1* KO mESCs, DNTTIP1 association with chromatin was strongly reduced in the absence of ELMSAN1 (*Figure 3D*). However, some DNTTIP1 enrichment in *Elmsan1* KO mESCs could still be observed, suggesting that DNTTIP1 retains the ability to bind to chromatin when ELMSAN1 is depleted (*Figure 3D*). To study the genome-wide MiDAC binding landscape in mESCs, we compared the occupancy patterns of DNTTIP1 and HDAC1. We found that many DNTTIP1-bound regions (58%) also displayed considerable enrichment for HDAC1, and that 40% of all HDAC1 peaks colocalized with DNTTIP1 (*Figure 3E*). Considering that HDAC1 also exists within several other complexes, including SIN3, NuRD, and CoREST, this finding suggests that a significant number of HDAC1-occupied loci is bound by MiDAC. Collectively, our findings suggest that MiDAC localizes genome-wide to enhancers and promoters in mESCs, thereby transcriptionally activating or inhibiting neurodevelopmental genes. We next examined activators and repressors of neurite outgrowth more specifically. We investigated SLIT3 and NETRIN1 (NTN1), which are key ligands of the SLIT/ROBO and NETRIN/DCC/UNC signaling pathways, and are regulators of neuron maturation and neurite outgrowth but are also playing roles in angiogenesis, lung morphogenesis, mammary gland development and cancer progression involving processes such as cell migration, cell interaction and cell adhesion (*Bashaw and Klein, 2010*; *Blockus and Chédotal, 2016*; *Lai Wing Sun et al., 2011*; *Seiradake et al., 2016*). Both *Slit3* and *Ntn1* mRNA levels were decreased in *Dnttip1* KO and *Elmsan1* KO versus WT mESCs and NE (*Figures 1G* and *2H*; *Supplementary file 2*). Furthermore, DNTTIP1 was directly bound to the promoters and putative intra- and intergenic enhancers of *Slit3* and *Ntn1* in mESCs and many of these gene regulatory elements were also co-bound by HDAC1 (*Figure 3F*; *Figure 3—figure supplement 1E*). Conversely, a repressor of neurite outgrowth and morphogenesis, *sprouty4* (*Spry4*), and an inhibitor of neural differentiation, *Id1*, were both transcriptionally upregulated in mESCs and NE upon loss of DNTTIP1 or ELMSAN1 (*Figures 1G* and *2H*; *Supplementary file 2; Alsina et al., 2012*; *Lyden et al., 1999*; *Nam and Benezra, 2009*). DNTTIP1 and HDAC1 co-occupied the promoters and putative enhancers of *Spry4* and *Id1* (*Figure 3G*; *Figure 3—figure supplement 1F*). In summary, these results indicate that MiDAC binds to regulatory regions of activators and repressors of neural differentiation and neurite outgrowth, thereby positively or negatively modulating their transcriptional output, which is in accordance with

the phenotypic defects observed in *Dnttip1* KO and *Elmsan1* KO neurons (*Figure 2B,E,F*). We next sought to identify potential histone substrates of MiDAC by screening for global changes of several histone acetylation marks in WT, *Dnttip1* KO, and *Elmsan1* KO mESCs. Of the histone acetylation marks tested, only H4K20ac was found to be consistently increased in *Dnttip1* KO and *Elmsan1* KO mESCs, suggesting that H4K20ac could be a major MiDAC substrate (*Figure 3H*). Interestingly, H4K20ac has been recently described as a repressive histone acetylation mark, and its targeting by MiDAC might explain why we observed a greater number of downregulated compared to upregulated genes in *Dnttip1* KO and *Elmsan1* KO mESCs versus WT mESCs (*Kaimori et al., 2016*).

## MiDAC directly targets positive and negative regulators of neurite outgrowth during neural differentiation

To further decipher the role of MiDAC in regulating select target genes during neurogenesis, we performed ChIP analysis of DNTTIP1, ELMSAN1, HDAC1 and the histone acetylation marks H3K27ac (active) and H4K20ac (repressive) in NE after 12 days of differentiation. We focused our attention on the four previously described neurodevelopmental genes– *Slit3*, *Ntn1*, *Spry4*, and *Id1*– that we had identified as being bound by MiDAC in mESCs and transcriptionally regulated in mESCs and NE amongst other target genes (*Figures 1G*, *2H* and *3F* and G; *Figure 3—figure supplement 1E and F*). In WT NE, DNTTIP1, ELMSAN1, and HDAC1 occupied the promoter regions and putative enhancer regions of all four genes, and MiDAC binding was abrogated in *Dnttip1* KO and *Elmsan1* KO NE (*Figure 4A–C and F–H*; *Figure 4—figure supplement 1A-C and F-H*). Interestingly, we observed gene-specific changes in the active histone mark H3K27ac and the repressive histone mark H4K20ac depending on whether a gene was upregulated or downregulated in *Dnttip1* KO and *Elmsan1* KO compared to WT mESCs and NE. For example, genes that were positively regulated by MiDAC in NE, such as *Slit3* and *Ntn1*, displayed a strong increase in H4K20ac in *Dnttip1* KO and *Elmsan1* KO versus WT NE on promoter and associated enhancer regions without major changes in H3K27ac (*Figure 4D and E*; *Figure 4—figure supplement 1D and E*). The reverse scenario was observed for genes that were repressed by MiDAC in NE cells, such as *Spry4* and *Id1*, which showed a higher enrichment of H3K27ac at their promoter and putative enhancer regions without any alteration in H4K20ac in *Dnttip1* KO and *Elmsan1* KO compared to WT NE (*Figure 4I and J*; *Figure 4—figure supplement 1I and J*). In summary, our data indicate that MiDAC differentially regulates the active H3K27ac and repressive H4K20ac marks on neurodevelopmental genes during neurogenesis. Furthermore, genes that are positively regulated by MiDAC tend to be activators of neurite outgrowth and morphogenesis, whereas genes that are under negative control of MiDAC appear to be repressors of neural differentiation.

## MiDAC regulates neurite outgrowth via the SLIT3/ROBO3 and NTN1/UNC5B signaling pathways

We noticed that several downregulated genes in *Dnttip1* KO and *Elmsan1* KO mESCs and NE encoded secreted ligands of the SLIT and NETRIN families, which are important regulators of neurite development and axon guidance signaling (*Figures 1G* and *2H*). Furthermore, our findings also indicated that MiDAC is bound to promoters and enhancers of *Slit3* and *Ntn1*, suggesting that MiDAC directly activates the genes encoding these secreted ligands during neurogenesis (*Figure 3F*; *Figure 3—figure supplement 1E*). However, MiDAC did not transcriptionally regulate the genes of their cognate receptors *Robo3* and *Unc5b* in mESCs and NE and even though it was bound to the promoter and putative enhancer regions of *Unc5b* in mESCs no enrichment for MiDAC components was detected on the promoters of *Robo3* and *Unc5b* in NE (*Figure 4—figure supplement 2A-F*). Thus, we hypothesized that the observed defect in neurite outgrowth upon loss of MiDAC might be caused at least in part by transcriptional downregulation of *Slit3* and/or *Ntn1*. To address this question, we differentiated WT, *Dnttip1* KO and *Elmsan1* KO mESCs into NE for 12 days and supplemented *Dnttip1* KO and *Elmsan1* KO cells daily with conditioned medium (CM) from differentiating WT cells from day 7 onward (*Figure 5—figure supplement 1A*). After 12 days of differentiation, NE was stained for MAP2 followed by analysis of neurite length (*Figure 5A*). Consistent with our previous findings, the average neurite length per neuron was reduced in *Dnttip1* KO and *Elmsan1* KO compared to WT neurons and was partly restored upon treatment with CM of WT NE (*Figure 5A and B*). Interestingly, classification of the neurite length into two categories, <50 µm or ≥50 µm,

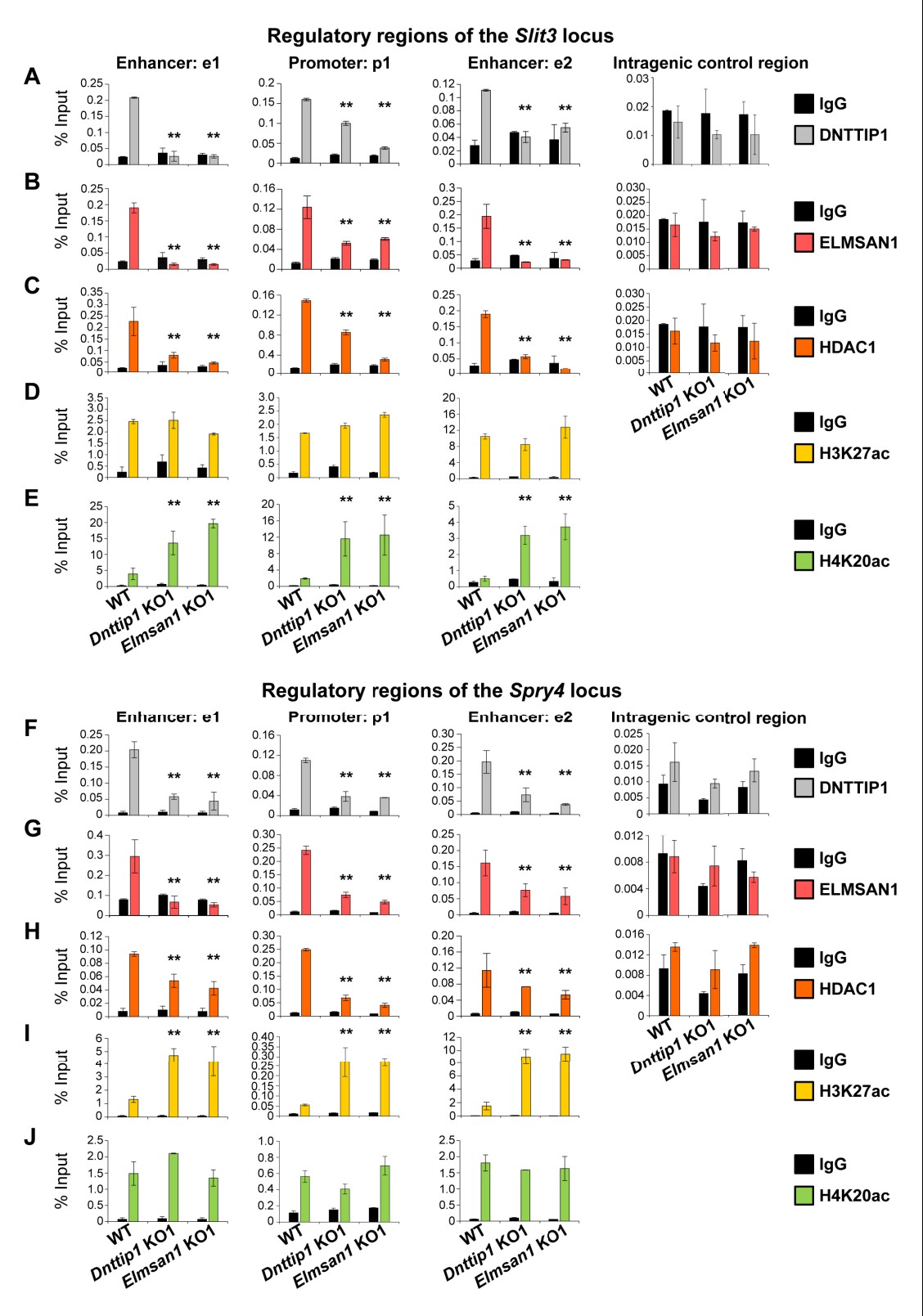

**Figure 4.** MiDAC directly targets positive and negative regulators of neurite outgrowth during neural differentiation. (A–J) qPCR from manual ChIP experiments against (A, F) DNTTIP1, (B, G) ELMSAN1, (C, H) HDAC1, (D, I) H3K27ac and (E, J) H4K20ac from WT, *Dnttip1* KO1 and *Elmsan1* KO1 NE targeting select promoter, putative enhancer and intragenic control regions of (A–E) *Slit3* or (F-J) *Spry4* loci as highlighted in *Figure 3F and G*. IgG was used as a control antibody. Unpaired t-test was performed throughout where **, p≤0.01; and ns, p>0.05 is not significant.

*Figure 4 continued on next page*

*Figure 4 continued*

The online version of this article includes the following figure supplement(s) for figure 4:

**Figure supplement 1.** MiDAC directly targets positive and negative regulators of neurite outgrowth during neural differentiation.
**Figure supplement 2.** MiDAC does not transcriptionally regulate the genes of the ROBO3 and UNC5B receptors of the SLIT3 and NTN1 signaling pathways.

showed that only longer neurites (≥50 μm in length) were significantly affected in *Dnttip1* KO and *Elmsan1* KO neurons and could be selectively rescued by supplementation with WT CM (*Figure 5B*). The impairment of longer neurites upon loss of MiDAC function implies that MiDAC might either regulate neuronal maturation or neurite outgrowth at an advanced stage. In addition, the reduced number of neurites per neuron in *Dnttip1* KO and *Elmsan1* KO versus WT neurons was also significantly restored upon treatment with CM of WT NE (*Figure 5C*). To further validate these findings, we used a chamber assay to co-culture granule neuron progenitor cells (GNPs) in the lower chamber in parallel with differentiating WT, *Dnttip1* KO or *Elmsan1* KO NE in the upper chamber comprising a time window from day 7–12 of differentiation to induce neuronal network formation of GNPs (*Figure 5—figure supplement 1B*). Co-culturing of GNPs with differentiating WT NE resulted in extensive neuronal network formation, while neuronal network formation was severely impaired when GNPs were co-cultured with *Dnttip1* KO and *Elmsan1* KO NE (*Figure 5—figure supplement 1C and*

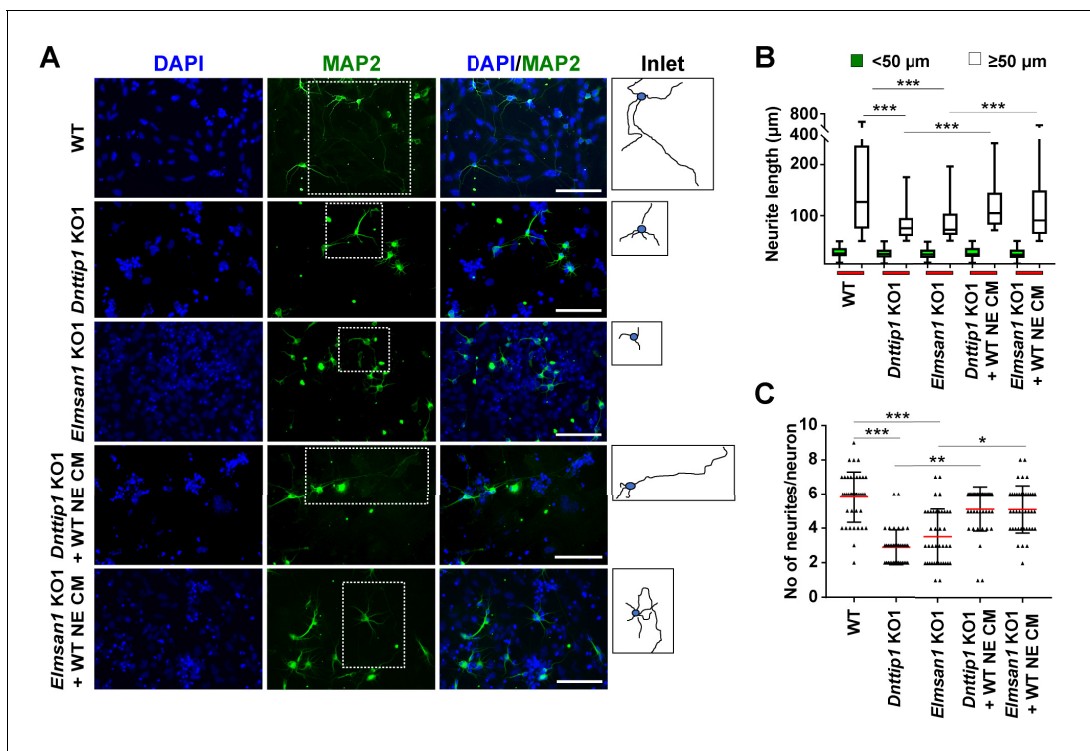

**Figure 5.** The regulatory role of MiDAC in neural differentiation and neurite outgrowth is carried out in part by secreted components. (A) MAP2 IF staining after 12 days of differentiation performed on WT, *Dnttip1* KO1 and *Elmsan1* KO1 NE supplemented daily with conditioned medium (CM) of WT NE from day 7–12. Nuclei were stained with DAPI. For analysis the neuronal cell body (blue) and its neurites (black) were manually traced with ImageJ software and for each sample one traced neuron is displayed in the inlet. The white scale bar represents 50 μm. (B, C) Quantification of (B) neurite length and (C) the total number of neurites per neuron from the MAP2 IF staining in (A) using ImageJ. (B) Neurite length was divided into two categories of short neurites <50 μm (green box plots) and longer neurites ≥50 μm (white box plots). (B, C) The neurites of 200 neurons were assessed per sample. One-way ANOVA was performed throughout where ***, p≤0.001; **, p≤0.01; and *, p≤0.05.

The online version of this article includes the following figure supplement(s) for figure 5:

**Figure supplement 1.** The regulatory role of MiDAC in neural differentiation and neurite outgrowth is carried out in part by secreted components.

*D*). In summary, these results suggest that the loss of MiDAC function in NE results in a lack of certain secretome components that are otherwise required for proper neurite outgrowth.

To further establish the specific effects of MiDAC on the SLIT3 and NTN1 signaling pathways, we investigated the ligand-receptor pairs SLIT3/ROBO3 and NTN1/UNC5B. Importantly, compared to lysates from WT NE, lysates from *Dnttip1* KO NE showed strong reductions in SLIT3 and NTN1, whereas ROBO3 and UNC5B were not significantly affected (*Figure 6A*). Furthermore, we also observed a significant reduction of SLIT3 and NTN1 in CM of *Dnttip1* KO versus WT NE (*Figure 6A*). The downstream effectors DAB1 of the NTN pathway and FAK of the SLIT/NTN signaling cascade were also deactivated, as shown by decreased phosphorylation of DAB1 and FAK in *Dnttip1* KO versus WT NE, despite no change in total protein levels (*Figure 6A*; *Zelina et al., 2014*). Having established that the SLIT3 and NTN1 signaling cascades are directly targeted by MiDAC during neurogenesis, we sought to test whether the loss of activity of these pathways is sufficient to explain the neurite outgrowth defects detected in neurons that had lost MiDAC function. To address this question, we differentiated *Dnttip1* KO mESCs into NE for 12 days and supplemented the CM from day 7 onward with recombinant SLIT3, NTN1, or a combination of SLIT3 and NTN1 (*Figure 5—figure supplement 1A*). To assess the specific effects of these ligands with their corresponding pathways, we combined the supplementation of CM with recombinant ligands with a blocking approach involving antibodies directed against the extracellular domains of either ROBO3 (SLIT3 pathway), UNC5B (NTN1 pathway), or a combination of ROBO3/UNC5B (*Figure 5—figure supplement 1A*). We found that SLIT3, NTN1, and SLIT3/NTN1 were all able to largely restore the neurite outgrowth defects observed in *Dnttip1* KO neurons and that the restoration of the phenotype was directly mediated by signaling through the ROBO3 and UNC5B signaling receptors (*Figure 6B–D*). We also obtained similar results using a modified version of the chamber assay described above, in which GNPs were treated with CM of WT or *Dnttip1* KO NE and identical combinations of recombinant ligands and blocking antibodies (*Figure 5—figure supplement 1B*). The reduced neuronal network formation, that was observed when GNPs were treated with CM of *Dnttip1* KO NE, was largely rescued in the presence of SLIT3, NTN1, or SLIT3/NTN1, and the restoration of network formation was significantly abrogated by pre-blocking GNP-derived neurons with neutralizing antibodies against ROBO3 and/or UNC5B before ligand-containing CM of *Dnttip1* KO NE was added (*Figure 6—figure supplement 1A and B*). Taken together, these findings indicate that the loss of MiDAC function in NE during neural differentiation results in strongly reduced SLIT3 and NTN1 signaling which accounts for the majority of the observed neurite outgrowth defects.

## Discussion

Here, we report that in mESCs, DNTTIP1 in association with ELMSAN1 and HDAC1, form a chromatin-associated mitotic deacetylase complex called MiDAC. We provide evidence that MiDAC is enriched at promoters and enhancers genome-wide and can function as an activator and repressor of transcription in mESCs. More specifically, we uncover a crucial role for MiDAC in regulating a neurodevelopmental gene expression program that controls neurite outgrowth and network formation during neurogenesis. MiDAC achieves this by positively regulating a set of neurodevelopmental genes including genes of the axon guidance ligands SLIT3 and NTN1, while at the same time suppressing negative regulators of neurogenesis such as *Spry4* and *Id1*. The results presented here support a model in which MiDAC is required to activate the promoters and enhancers of pro-neural genes such as *Slit3* and *Ntn1* by H4K20 deacetylation while in parallel repressing negative regulators of neurogenesis by removing H3K27ac from promoters and enhancers of genes such as *Spry4* and *Id1* to allow neurons to attain their required neurite length during differentiation (*Figure 7*).

MiDAC originally obtained its name from a chemoproteomic profiling study due to its increased association with certain HDAC inhibitors in protein extracts from mitotically arrested versus non-synchronized proliferating cells that otherwise displayed comparable levels of DNTTIP1 (*Bantscheff et al., 2011*). Individual MiDAC subunits have been shown to interact with CCNA2 and/or the cyclin dependent kinase CDK2, and were reported to positively regulate the cell cycle inhibitors CDKN1A and CDKN1B and cell growth in various cancer cell lines, thus implying a potential role for MiDAC in cell cycle regulation (*Gizard et al., 2005*; *Gizard et al., 2006*; *Hein et al., 2015*; *Huttlin et al., 2015*; *Pagliuca et al., 2011*; *Sawai et al., 2018*; *Zhang et al., 2018*). However, we did not observe any effect on cell proliferation, cell cycle distribution, or any transcriptional changes

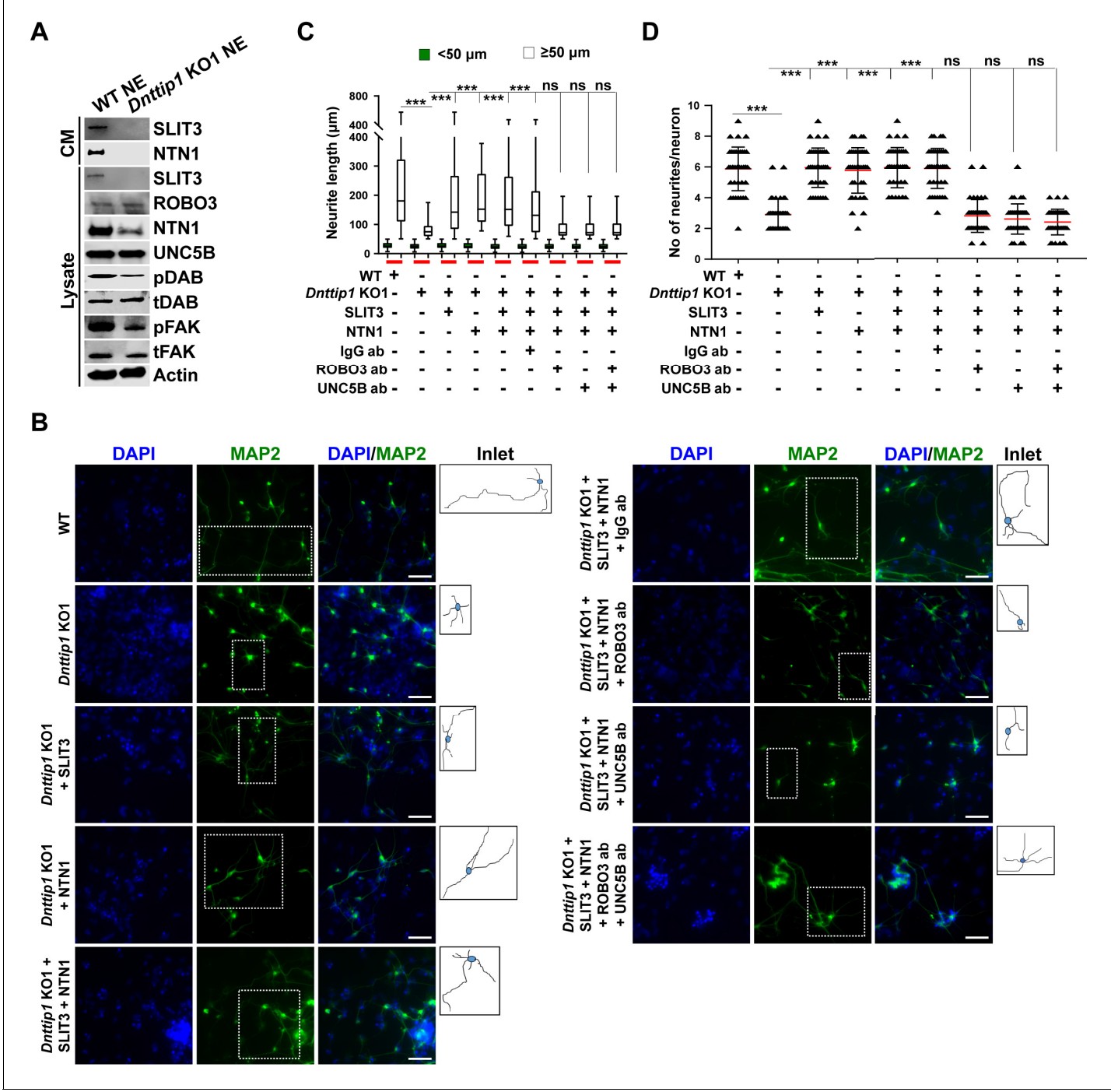

**Figure 6.** MiDAC regulates neurite outgrowth via the SLIT3/ROBO3 and NTN1/UNC5B signaling pathways. (A) WB for signaling components of the SLIT3/ROBO3 and NTN1/DCC/UNC5B signaling axes from CM and total cell lysates of WT and *Dnttip1* KO1 NE after 12 days of differentiation. To enrich SLIT3 and NTN1 from CM, IPs were performed with SLIT3 and NTN1 antibodies from CM of WT and *Dnttip1* KO1 NE. Actin is the loading control for the total cell lysates. (B) Assay to rescue the neurite outgrowth defects in *Dnttip1* KO1 NE. CM of *Dnttip1* KO1 NE was supplemented with the recombinant signaling ligands SLIT3 and/or NTN1 from day 7–12 without or with preblocking of *Dnttip1* KO1 NE with IgG or signaling receptor antibodies against ROBO3 and/or UNC5B. MAP2 IF staining was performed after 12 days of differentiation. Nuclei were stained with DAPI. To facilitate analysis the neuronal cell body (blue) and its neurites were manually traced with ImageJ software and for each sample one traced neuron is displayed in the inlet. The white scale bar represents 50 μm. (C, D) Quantification of (C) neurite length and (D) the total number of neurites per neuron from the MAP2 IF staining in (B) using ImageJ. (C) Neurite length was divided into two categories of short neurites <50 μm (green box plots) and longer neurites ≥50 μm (white box plots). (C, D) The neurites of 200 neurons were assessed per sample. One-way ANOVA was performed throughout where ***, p≤0.001; and ns, p>0.05 is not significant.

*Figure 6 continued on next page*

*Figure 6 continued*

The online version of this article includes the following figure supplement(s) for figure 6:

**Figure supplement 1.** MiDAC regulates neurite outgrowth via the SLIT3/ROBO3 and NTN1/UNC5B signaling pathways.

in the above-mentioned cell cycle-related genes (*Ccna2*, *Cdkn1a*, *Cdkn1b*) in mESCs with a loss in MiDAC function, arguing in favor of a cell- and/or context-specific role for MiDAC in controlling cell proliferation and the cell cycle.

The C-terminus of DNTTIP1 has been reported to bind to a specific DNA binding motif following ChIP-seq studies utilizing a FLAG-tagged version of DNTTIP1 in HEK293 cells (motif: TGCAGTG-(14 bp)-CACTGCA flanked by AT-tracts) (*Koiwai et al., 2015*). In concordance with these results, our study confirms a chromatin-associated function and binding of DNTTIP1 to both promoter-proximal and -distal elements in mESCs. Although we did not find evidence of a clear DNTTIP1 consensus binding motif, we did detect a significant enrichment of several TF motifs at DNTTIP1-bound sites (data not shown). This implies that MiDAC is specifically recruited to chromatin by TFs and that its ability to bind nucleosomes might aid in MiDAC spreading from its initial recruitment site to carry out histone deacetylation. Furthermore, motif analysis of differentially expressed genes that display DNTTIP1 enrichment at their promoters or enhancers revealed the binding motifs of several TFs with previously reported roles in neurogenesis. We postulate that MiDAC upregulates a transcriptional program that is driven by ELK1, a member of the ETS family of pioneer transcription factors, while downregulating genes through RBFOX2, which negatively affects neurodevelopmental processes (*Besnard et al., 2011*; *Gehman et al., 2012*). However, it remains to be determined whether these

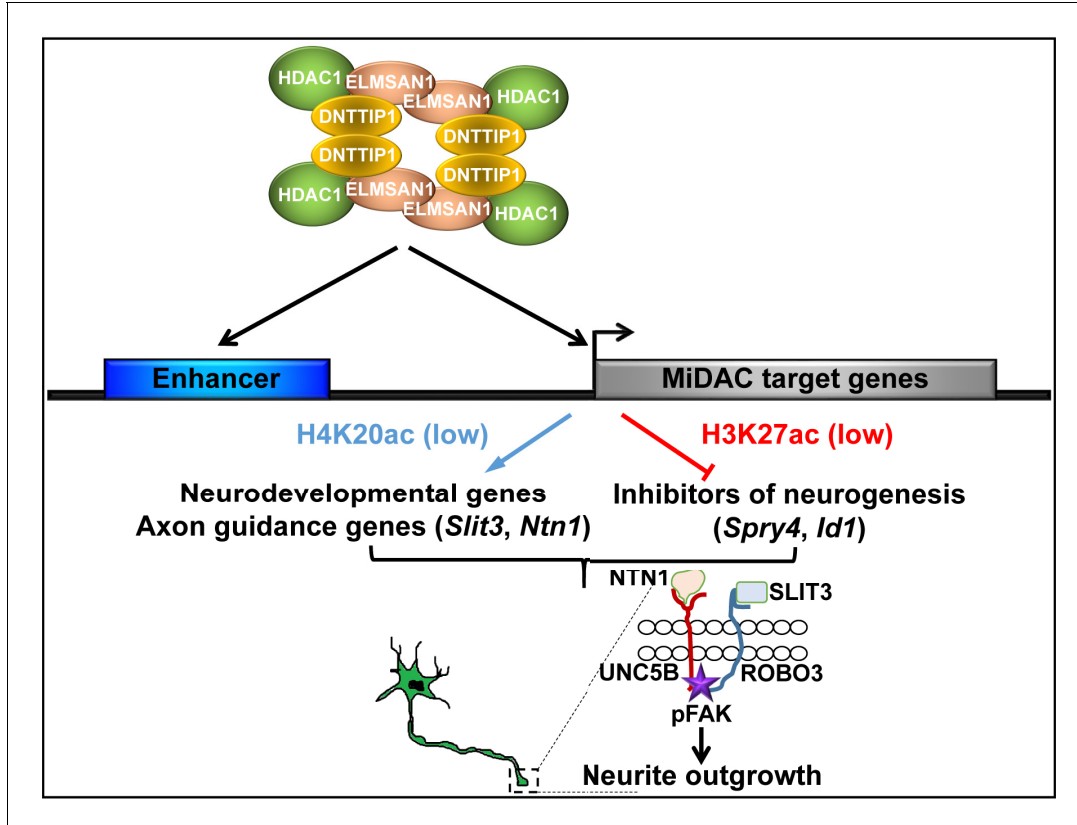

**Figure 7.** Model of MiDAC function in neurite outgrowth and morphogenesis. MiDAC directly binds to and deacetylates H4K20ac on regulatory elements of pro-neural genes such as those of the axon guidance ligands SLIT3 and NTN1 resulting in the activation of these genes. Conversely, MiDAC inhibits the gene expression of negative regulators of neurogenesis such as SPRY4 and ID1 by binding and removing H3K27ac from their promoters and enhancers. SLIT3 and NTN1, the downstream targets of MiDAC, bind to their cognate receptors ROBO3 and UNC5B respectively thereby activating the signaling cascade responsible for promoting neurite outgrowth.

TFs affect MiDAC recruitment and/or function during neurogenesis. Although DNTTIP1 levels are strongly reduced in *Elmsan1* KO mESCs, we still observed some DNTTIP1 recruitment to chromatin. With TRERF1 and ZFP541 (ZNF541 in humans) two ELMSAN1-related proteins have been described that could also potentially be incorporated into MiDAC (*Choi et al., 2008*; *Gizard et al., 2001*). While ZFP541 is expressed in a tissue-specific manner during spermatogenesis and not in mESCs, TRERF1 expression can be detected in mESCs (data not shown) (*Choi et al., 2008*). Thus, the remaining DNTTIP1 in *Elmsan1* KO mESCs might form an alternative MiDAC complex with the ELMSAN1 paralog TRERF1 which could account for the residually detected DNTTIP1 recruitment in the absence of ELMSAN1.

HDAC1/2-containing complexes are generally considered repressors of transcription (*Kadosh and Struhl, 1998*; *Rundlett et al., 1998*; *Yang et al., 1996*). However, accumulating evidence suggests that HDAC1/2 might also be involved in transcriptional activation (*Bernstein et al., 2000*; *Harrison et al., 2011*; *Yamaguchi et al., 2010*). Moreover, genome-wide studies have shown that HDAC1/2 localize to promoters, gene bodies, and enhancers of actively transcribed genes, thus implicating HDAC1/2 as positive regulators of transcription (*Wang et al., 2009*). Here, our finding that a high percentage of DNTTIP1-bound loci is associated with both down- and upregulated DEGs in *Dnttip1* KO versus WT mESCs and NE shows for the first time a direct role for MiDAC in both transcriptional activation and repression. The exact mode of action whereby MiDAC achieves activation or repression has yet to be elucidated in greater detail, but our studies provide evidence for at least two possible scenarios. First, MiDAC could control gene activation and repression by deacetylating histones. Most histone acetylation marks, including H3K27ac, correlate with transcriptional activation, but recently H4K20ac has been reported as a mark that is associated with silenced genes (*Kaimori et al., 2016*). Our data indicate that MiDAC might function as a repressor of genes that are marked by H3K27ac and more broadly as an activator on repressed genes by mediating deacetylation of H4K20. However, the relationship between H4K20ac and MiDAC in regulating gene expression is currently only restricted to correlative evidence and it remains to be determined to which degree H4K20ac is instructive in mediating gene repression and whether this mark constitutes a direct substrate of MiDAC. As a second possibility, MiDAC might exert its function by deacetylating TFs or other chromatin-modifying proteins such as histone acetyltransferases (HATs). If specific TFs or HATs are negatively regulated by deacetylation through MiDAC at distinct target sites, this could also potentially explain the increase in H3K27ac (through CBP/p300) or other histone acetylation marks on enhancers and promoters of up- or downregulated genes between WT, *Dnttip1* KO and *Elmsan1* KO mESCs and NE.

HDAC1 and HDAC2 are important regulators of many developmental processes including neurogenesis (*Kelly and Cowley, 2013*). But their function remains enigmatic as they can be recruited into at least four independent HDAC1/2-containing complexes (*Millard et al., 2017*). While the HDAC1/2-containing SIN3, NuRD and CoREST complexes have been implicated in neurogenesis, the physiological function of MiDAC has been unexplored to date (*Andrés et al., 1999*; *Knock et al., 2015*; *Nitarska et al., 2016*; *Wang et al., 2016*). Here, we report that MiDAC directly controls a neurodevelopmental gene expression program and specifically targets enhancers and promoters of positive and negative regulators that are involved in neurite outgrowth during neural differentiation. While this study emphasizes the role of MiDAC in neural differentiation and neurite outgrowth, it is evident from our gene expression analyses that this neurodevelopmental gene expression program only constitutes a fraction of the MiDAC-regulated transcriptome in mESCs pointing to other biological roles of MiDAC. SLIT/ROBO, EPHRIN/EPH, NETRIN/DCC/UNC, and SEMAPHORIN/PLEXIN signaling represent the four major canonical axon guidance pathways (*Bashaw and Klein, 2010*). Integration of our gene expression and ChIP data showed that MiDAC directly activates the genes of the axon guidance ligands SLIT3 and NTN1 while repressing the genes of the negative regulators SPRY4 and ID1. SLIT3 and NTN1 are ligands for their cognate receptors ROBO3 and UNC5B, respectively. Recombinant SLIT3 and NTN1 either alone or in combination were able to largely rescue the neurite outgrowth defects in neurons that have lost MiDAC subunits. Interestingly, blocking only ROBO3 or only UNC5B signaling was sufficient to inhibit the rescue effects of SLIT3 in combination with NTN1, suggesting that a heteromeric ROBO3/UNC5B axis is required for efficient transduction of SLIT3/NTN1 signaling to ensure proper neurite outgrowth. While evidence exists that heteromeric receptor association and signaling of ROBO3/UNC5B occurs, it has yet to be determined whether this also applies to the role of MiDAC in an in

vivo system (*Zelina et al., 2014*). In summary, while individual TFs have been reported to transcriptionally regulate select axon guidance ligands or receptors little is known about their regulation through chromatin-modifying components (*Kim et al., 2016*; *Labrador et al., 2005*). Here, we describe how the axon guidance ligands SLIT3 and NTN1 are transcriptionally controlled by an epigenetic regulator, the MiDAC complex. Finally, neurite outgrowth defects are observed in several neurodevelopmental disorders such as autism, epilepsy, fragile X-syndrome and psychiatric disorders (*Gilbert and Man, 2017*; *Huang and Song, 2019*; *Krejčí et al., 2017*; *Wen et al., 2017*). However, the underlying molecular mechanisms remain unclear. Our study implicates MiDAC as an essential regulator of neurite outgrowth which suggests that it might be playing an important role in certain neurodevelopmental disorders. Interestingly, a recent study reported noncoding mutations in several MiDAC components among autism patients, thus supporting a potential role for MiDAC in certain neurodevelopmental disorders (*Zhou et al., 2019*). Future studies of MiDAC in this context are thus likely to yield insightful mechanistic details about disease pathogenesis.

## Materials and methods

### Cell lines

Male C57/BL6 Bruce-4 mESCs were purchased from Millipore (Millipore, SF-CMTI-2). WT mESCs and their *Dnttip1* KO and *Elmsan1* KO derivatives were originally cultured under feeder free culture conditions (KnockOut DMEM medium) on 0.1% gelatin-coated flasks or plates. After CRISPR/Cas9 genome-editing mESCs were transitioned to chemically defined naïve culture conditions (2iL) on 0.1% gelatin-coated flasks or plates. mESC identity was authenticated by various methods including alkaline phosphatase staining, staining and cytometry analysis of the pluripotency marker FUT4 (SSEA-1) and the ability to differentiate into the three main germ layers. All mESC lines tested negative for mycoplasma contamination.

Feeder-free culture conditions (SL):
KnockOut DMEM (no L-Glutamine); 15% FBS (ES cell qualified); 2 mM L-Glutamine or GlutMAX (1:100); 1 mM Sodium Pyruvate (1:100); 0.1 mM MEM Non-Essential Amino Acids (1:100); Embryomax Nucleosides (1:100); 0.1 mM β-Mercaptoethanol (1:500); 1000 U/ml LIF (1:10000 from $10^7$ U/ml stock); 50 U/ml Pen/Strep (1:100).
Chemically defined naïve culture conditions (2iL):
50% DMEM/F-12; 50% Neurobasal Medium (no L-Glutamine); B-27 Supplement, minus vitamin A (1:100); N-2 Supplement (1:200); 2 mM L-Glutamine or GlutaMAX (1:100); 0.1 mM β-Mercaptoethanol (1:500); 3 μM CHIR 99021 (GSK3β inhibitor) (1:1000 from 3 mM stock in DMSO); 1 μM PD 0325901 (MEK inhibitor) (1:1000 from 1 mM stock in DMSO); 1000 U/ml LIF (1:10000 from $10^7$ U/ml stock); 50 U/ml Pen/Strep (1:100).

### Neural differentiation

#### Directed differentiation into neuro-ectoderm

WT, *Dnttip1* KO and *Elmsan1* KO mESCs were dissociated by trypsinization, resuspended in differentiation medium 1 and pelleted by centrifugation. $10^6$ mESCs were resuspended in 3 ml differentiation medium 1 and then transferred into one well of 6-well ultra-low attachment plate (Corning, 3471). Cells were grown in differentiation medium 1 for 3 days. Differentiation medium 1 was changed every day. After 3 days the emerging embryoid bodies (EBs) were cultured in differentiation medium 2 containing 2 μM retinoic acid (Santa Cruz Biotechnology, sc-200898) for an additional 3 days. Differentiation medium 2 was changed every day. After 6 days the EBs were picked up gently with a 200 μl large orifice tip and transferred into a 1.5 ml tube. The EBs were allowed to settle down by gravity followed by careful aspiration of the supernatant. 1 ml differentiation medium 3 containing 20 ng/ml FGF2 and 20 ng/ml EGF was carefully added to the 1.5 ml tube containing the EBs, and the EBs were gently picked up with a 200 μl large orifice tip and transferred onto a 0.1% gelatin-coated well of a standard 6-well plate (Corning, 3516). Subsequently, 1 ml of additional differentiation medium 3 was carefully added to the well containing the EBs. The EBs were allowed to differentiate for the next 6 days, with daily changing of differentiation medium 3. The EBs will attach slowly over time and aster-shaped neurons will emerge from the periphery of the EBs forming increasingly longer neurite extensions and networks. Neurite outgrowth was quantified and

visualized after 12 days of differentiation. Experiments to rescue the neurite outgrowth defects of *Dnttip1* KO and *Elmsan1* KO neurons with CM from WT NE were conducted as above with the following alterations. After 6 days of differentiation all *Dnttip1* KO and *Elmsan1* KO CM was removed and replaced daily with 50% fresh differentiation medium 3 and 50% CM from WT NE for the next 6 days. Experiments to rescue the neurite outgrowth defects of *Dnttip1* KO neurons with axon guidance ligands were carried out as described above except that recombinant SLIT3 (R&D Systems, 9296-SL-050), NTN1 (R&D Systems, 6419-N1-025) or a combination of SLIT3/NTN1 were added daily to differentiation medium 3 (at a concentration of 0.5 and 0.25 µg/ml for SLIT3 and NTN1 respectively) from day 7 of differentiation onward. The rescued neurite outgrowth defects of *Dnttip1* KO neurons via SLIT3, NTN1 or SLIT3/NTN1 supplementation were blocked by using antibodies directed against the extracellular domain of the axon guidance receptors ROBO3 (R&D Systems, AF3155), UNC5B (R&D Systems, MAB1006), a combination of ROBO3/UNC5B or IgG (Millipore, 12–370) as a negative control. Blocking was performed daily from day 7 onward by adding 0.5 µg of each antibody to differentiation medium 3 two hours before addition of the ligands SLIT3, NTN1 or SLIT3/NTN1.

> Differentiation medium 1:
>     50% DMEM/F-12; 50% Neurobasal Medium (no L-Glutamine); B-27 Supplement with vitamin A, serum free (1:100); N-2 Supplement (1:200); 0.1 mM β-Mercaptoethanol (1:500); 50 U/ml Pen/Strep (1:100).
> Differentiation medium 2:
>     Differentiation medium 1; 2 µM retinoic acid.
> Differentiation medium 3:
>     Differentiation medium 1; 20 ng/ml FGF2; 20 ng/ml EGF.

## Chamber assay to assess neuronal network formation from granule neuron progenitor cells (GNPs)

GNPs were isolated from mouse cerebella from postnatal day 6–7 (a gift from Martine Roussel) (*Vo et al., 2016*). GNPs were grown as neurospheres in GNP medium in 6-well ultra-low attachment plates (Corning, 3471). GNP neurospheres were dissociated by trypsinization, resuspended in GNP culture medium and pelleted by centrifugation. $10^5$ GNPs were resuspended in 600 µl GNP culture medium and seeded into one well of a 24-well ultra-low attachment plate (Corning, 3473) and allowed to form neurospheres for 3 days. GNP culture medium was changed every day. From this point onward GNPs were differentiated in the same fashion as described in 'Directed differentiation into neuro-ectoderm' by sequential culturing in differentiation medium 2 and 3 and adjustment of media volumes to a 24-well format (one fifth of a 6-well format) with the following alterations. After culturing in differentiation medium 2 GNPs were plated into one well of a 24-well transwell chamber (pre-coated with 1% gelatin) in differentiation medium 3 (Corning, 354480) and differentiated for another 6 days. WT, *Dnttip1* KO and *Elmsan1* KO mESCs were differentiated according to the 'Directed differentiation into neuro-ectoderm' protocol for the first 6 days but experiments were scaled down so that EBs were grown in one well of a 24-well ultra-low attachment plate (Corning, 3473) and all media volumes were adjusted to one fifth of the 6-well protocol. The EBs were carefully transferred into a matrigel-coated insert and allowed to differentiate for the next 6 days, with daily changing of differentiation medium 3 in the insert and GNP medium in the bottom chamber. In parallel, GNP medium and MEF CM was added to separate inserts as positive and negative controls, respectively and media changes were carried out in the same way as for differentiation medium 3. Experiments to rescue the network formation defects of GNP-derived neurons caused by culturing in CM of *Dnttip1* KO NE were carried out as described above for the first 6 days of differentiation. However, instead of co-culturing GNPs with *Dnttip1* KO NE, GNPs were supplemented daily with CM of *Dnttip1* KO NE (50%), 50% fresh differentiation medium 3 and recombinant SLIT3 (R&D Systems, 9296-SL-050), NTN1 (R&D Systems, 6419-N1-025) or a combination of SLIT3/NTN1 for 6 days from the day of GNP seeding. Network formation of GNP-derived neurons grown in CM of *Dnttip1* KO NE and differentiation medium 3 supplemented with SLIT3, NTN1 or SLIT3/NTN1 was blocked by using antibodies directed against the extracellular domain of the axon guidance receptors ROBO3 (R&D Systems, AF3155), UNC5B (R&D Systems, MAB1006), a combination of ROBO3/UNC5B or IgG (Millipore, 12–370) as a negative control. Blocking was performed daily for 6 days

from the day of GNP seeding by aspirating the old medium and adding 0.1 µg of each blocking antibody in fresh differentiation medium 3 two hours before adding CM of *Dnttip1* KO NE (50%), 50% fresh differentiation medium 3 with either the ligands SLIT3, NTN1 or SLIT3/NTN1. Network formation of GNP-derived mature neurons was quantified and visualized 6 days after GNPs were seeded. A neuronal network was scored when all neurite projections of an individual neuron were well connected with the neurites of neighboring neurons and formed a closed local circuit as assessed by TUBB3 staining (*Doetsch and Alvarez-Buylla, 1996*; *Shepherd, 1998*). The percentage of network formation was scored as follows: =total number of complete networks per TUBB3-positive neuron/ total number of DAPI-positive cells) x 100. The different groups were statistically analyzed using ONE-way ANOVA t-test.

GNP medium:
Neurobasal medium (no L-Glutamine); 2 mM L-Glutamine; B-27 Supplement with vitamin A, serum free (1:100); N-2 Supplement (1:200); 20 ng/ml FGF2; 20 ng/ml EGF; 50 U/ml Pen/ Strep (1:100).

## Alkaline Phosphatase staining and quantification

Alkaline phosphatase staining of mESCs was carried out with the Alkaline Phosphatase Detection Kit (Millipore, SCR004) following the manufacturer's instructions. 100 mESC colonies were assessed per genotype.

## ChIP-seq library preparation and sequencing

DNA was quantified using the Quant-iT PicoGreen dsDNA Assay (Thermo Fisher Scientific, P11496). Libraries were prepared with the KAPA HyperPrep Library Kit (Roche, 07962363001) and analyzed for insert size distribution with the High Sensitivity DNA Kit (Agilent, 5067–4626) on a 2100 Bioanalyzer or the High Sensitivity D1000 ScreenTape Assay (Agilent, 5067–5584, 5067–5585, 5067–5587, 5067–5603) on a 4200 TapeStation. Libraries were quantified using the Quant-iT PicoGreen dsDNA Assay. Single end 50 cycle sequencing was performed on a HiSeq 2500, HiSeq 4000, or NovaSeq 6000 System (all from Illumina).

## Chromatin immunoprecipitation (ChIP)

ChIPs were performed according to a modified version of *Lee et al. (2006)*. For ChIP-seq applications of non-histone proteins ChIP was carried out with $5 \times 10^7$ mESCs and for histone ChIPs with $2$–$2.5 \times 10^7$ mESCs. For manual ChIP applications of non-histone proteins ChIP was carried out with $2 \times 10^6$ NE cells and for histone ChIPs with $10^6$ NE cells. For non-histone ChIPs dual crosslinking was performed at room temperature (RT) for 30 min with 2 mM disuccinimidyl glutarate (DSG) in DPBS followed by addition of paraformaldehyde to a final concentration of 1% and further crosslinking for 15 min. For histone ChIPs crosslinking was performed at RT for 15 min with 1% paraformaldehyde. Crosslinking was quenched with 150 mM glycine for 5 min at RT. After quenching the cells were pelleted, the supernatant aspirated and the cell pellet washed once in DPBS and then snap-frozen and stored at −80°C or immediately further processed. The crosslinked cell pellet was carefully resuspended in Lysis Buffer 1 with a transfer pipet and the cell suspension incubated for 10 min on a nutator at 4°C. Nuclei were then pelleted and the supernatant aspirated. The pelleted nuclei were carefully resuspended in Lysis Buffer 2 with a transfer pipet and the cell suspension incubated for 10 min on a nutator at 4°C. Nuclei were then pelleted again and the supernatant aspirated. The nuclear pellet was then resuspended in Lysis Buffer 3 and the suspension sonicated with a probe sonicator (Fisher Scientific Model 705 Sonic Dismembrator) at output setting 55 (27–33 W) for 12 cycles with each cycle constituting a 30 sec sonication burst followed by a 60 sec pause. After sonication 1/20 volume of 20% Triton X-100 was added and the sample was mixed. The chromatin was cleared by centrifugation for 10 min at 20000 g at 4°C and the supernatant was transferred to a fresh tube. The cleared chromatin was then stored at 4°C. For each ChIP 100 µl Protein A/G-Plus agarose (Santa Cruz Biotechnology, sc-2003) slurry was used. The beads were washed in 1 ml Blocking Solution, pelleted and the supernatant was removed. The bead wash step was repeated another time. The beads were resuspended in 1 ml Blocking Solution followed by addition of the appropriate antibody and were incubated on a nutator at 4°C overnight. For ChIP-seq applications of non-histone proteins and histones 10 µg of antibody was used. For manual ChIP applications of non-histone proteins and

histones 3 µg of antibody was used. On the next day the stored chromatin was cleared another time by centrifugation for 10 min at 20000 g at 4°C and the supernatant was transferred to a fresh tube. A small chromatin aliquot of 50 µl was reserved as input control and was kept on ice until further processing (see below). The beads were washed twice with 1 ml Blocking Solution as on the previous day and resuspended in 100 µl Blocking Solution. The cleared chromatin was added and incubated with the beads on a nutator for 3 hr at 4°C. After the incubation the beads were pelleted and the supernatant aspirated. 1 ml Wash Buffer was added and the sample was mixed gently by inverting the tube several (8-10) times to ensure that all beads were fully resuspended. The beads were then pelleted and the supernatant aspirated. This wash step was repeated another four times (five washes total). A final wash with 1 ml TE buffer (10 mM Tris HCl pH 8.0; 1 mM EDTA) was added (including mixing, pelleting of beads and aspiration of supernatant). Chromatin was eluted with 200 µl Elution Buffer and the samples were incubated with shaking at 900 rpm for 30 min at 65°C in a ThermoMixer (Eppendorf, 5382000023 with a ThermoTop (Eppendorf, 5308000003). The beads were then pelleted and the supernatant transferred to a fresh tube. The input sample was brought up to 200 µl with Elution Buffer and all samples (input and ChIP samples) incubated overnight at 65°C (in a ThermoMixer with a ThermoTop) to reverse crosslinks. On the next day 200 µl TE and 8 µl of 10 mg/ml RNaseA were added followed by incubation in a ThermoMixer with a ThermoTop for 1 hr at 37°C. Next, 10 µl 20 mg/ml Proteinase K was added and the sample was incubated for an additional 2 hr at 55°C in a ThermoMixer with a ThermoTop. DNA was purified following the instructions of the QIAquick PCR Purification Kit from Qiagen (Qiagen, 28106). Elution of DNA was performed with 50 µl of prewarmed (55°C) EB buffer.

Lysis Buffer 1:
> 50 mM HEPES KOH pH 7.5; 140 mM NaCl; 1 mM EDTA; 10% Glycerol; 0.5% NP-40 (Igepal CA-630); 0.25% Triton X-100; protease inhibitors (Sigma, P8340) (1:200).

Lysis Buffer 2:
> 10 mM Tris HCl pH 8.0; 200 mM NaCl; 1 mM EDTA; 0.5 mM EGTA; protease inhibitors (Sigma, P8340) (1:200).

Lysis Buffer 3:
> 10 mM Tris HCl pH 8.0; 100 mM NaCl; 1 mM EDTA; 0.5 mM EGTA; 0.1% Na-deoxycholate; 0.5% N-lauroylsarcosine; protease inhibitors (Sigma, P8340) (1:200).

Blocking Solution:
> DPBS (Thermo Fisher Scientific, 14190144); 0.5% BSA (Sigma, A7906); 0.1% Triton X-100; protease inhibitors (Sigma, P8340) (1:500).

Wash Buffer:
> 50 mM HEPES KOH pH 7.5; 500 mM LiCl; 1 mM EDTA; 1% NP-40 (Igepal CA-630); 0.7% Na-deoxycholate; protease inhibitors (Sigma, P8340) (1:500).

Elution Buffer:
> 50 mM Tris HCl pH 8.0; 10 mM EDTA; 1% SDS.

## Flow cytometry

mESC colonies were dissociated by trypsinization, resuspended in 2iL medium and pelleted by centrifugation. Cells were washed once in DPBS followed by aspiration of the supernatant. While the cells were gently vortexed to avoid clumping, fixation was performed by dropwise addition of cold 70% ethanol. After fixation for 30 min on ice, the cells were pelleted by centrifugation and the supernatant aspirated. The cell pellet was then washed in DPBS and pelleted by centrifugation followed by aspiration of the supernatant. The cell pellet was resuspended in 50 µl of a 100 µg/ml RNase solution to ensure that only DNA, not RNA, is stained. 200 µl propidium iodide (PI) from a 50 µg/ml stock solution was added to each sample to stain the DNA. Cell cycle analysis of mESCs was carried out on a BD LRPFortessa cell analyzer. For cytometry analysis of PAX6-positive neural progenitor cells and TUBB3-positive neurons from 8 and 12 day old NE respectively, PAX6/DAPI and TUBB3/DAPI IF staining of NE was carried out as described under 'Immunofluorescence (IF)' followed by trypsinization and FACS analysis. PAX6- and TUBB3-positive cells were sorted with a BD FACSAria III cell sorter and the DNA content of individual cells was determined as a readout of DAPI intensity. Bar plots were generated from the raw data. Experiments were conducted in triplicate for cell cycle analysis of mESCs.

## Immunofluorescence (IF)

Neural differentiation was carried out in wells of a 6-well or 24-well plate as described under 'Neural differentiation'. Fixation Solution containing 8% paraformaldehyde was prepared from 16% paraformaldehyde (Electron Microscopy Sciences, 15710) and 10 x DPBS (Thermo Fisher Scientific, 14200075) and adjusted with $H_2O$ to achieve a final concentration of 1 x DPBS. 1 ml/200 µl (6-well/24-well) Fixation Solution was added to 1 ml/200 µl (6-well/24-well) of remaining culture medium and cells were incubated for 20 min at RT. The fixative was aspirated and rinsed with 2 ml/400 µl (6-well/24-well) DPBS (Thermo Fisher Scientific, 14190144) and the DPBS aspirated again. Permeabilization was performed with 1 ml/200 µl (6-well/24-well) Permeabilization Solution for 10 min at RT. Following aspiration of the Permeabilization Solution the cells were rinsed three times with 1 ml Wash Buffer and aspiration of the Wash Buffer after each rinse. Blocking was performed with 1 ml/200 µl (6-well/24-well) Blocking Buffer for one hour at RT. Blocking Buffer was aspirated and 500 µl/100 µl (6-well/24-well) Wash Buffer containing the primary antibody was added and the sample was incubated in a humidified light-tight chamber at 4°C overnight. On the next day Wash Buffer containing the primary antibody was aspirated. The sample was then washed three times for 5 min with 2 ml/400 µl (6-well/24-well) Wash Buffer with gentle shaking on a rotator at RT and aspiration of the Wash Buffer after each step. 500 µl/100 µl (6-well/24-well) Wash Buffer containing the fluorochrome-conjugated secondary antibody and DAPI (1 µg/ml final concentration) was added and the sample was incubated with gentle shaking on a rotator in a humidified light-tight chamber for one hour at RT. The sample was again washed three times for 5 min with 2 ml/400 µl (6-well/24-well) Wash Buffer with gentle shaking on a rotator at RT and aspiration of the Wash Buffer after each step. 1 ml/200 µl (6-well/24-well) DPBS was added and the sample was imaged with an inverted widefield microscope platform (Leica, DMi8).

> Antibody dilutions for IF:
> Mouse α-MAP2 (Sigma, M9942); 1:1000. Mouse α-TUBB3 (BioLegend, 801201); 1:1000.
> Fixation Solution:
> DPBS; 8% paraformaldehyde.
> Permeabilization Solution:
> DPBS; 0.2% Triton X-100.
> Blocking Buffer:
> DPBS; 50 mg/ml BSA (5%); 0.1% goat serum.
> Wash Buffer:
> DPBS; 5 mg/ml BSA (0.5%).

## Immunoprecipation (IP)

$2 \times 10^7$ mESCs were used for each IP. mESCs were scraped off with a cell scraper and pelleted by centrifugation. The cell pellet was resuspended in DPBS and pelleted again. Cytoplasmic lysis was performed in 1 ml Lysis Buffer 1 by resuspending the cell pellet followed by incubation on a nutator for 5 min at 4°C. Nuclei were centrifuged for 5 min at 1500 g at 4°C, the supernatant (cytoplasmic fraction) aspirated and the nuclear pellet resuspended in 500 µl Lysis Buffer 2. The protein concentrations of the nuclear extracts were determined by a protein assay (BioRad, 5000006) with a spectrophotometer (Eppendorf, 6133000010) using disposable cuvettes (Eppendorf, 0030079353). Equal protein amounts were used for all samples of the same IP experiment and volumes were adjusted to 545 µl with Lysis Buffer 2. 45 µl of each sample was removed and used as Input. For each IP 50 µl Dynabeads Protein G (Thermo Fisher Scientific, 10004D) slurry was used. The beads were washed in 1 ml Lysis Buffer 2, collected with a magnetic stand (Thermo Fisher Scientific, 12321D) and the supernatant was removed. The bead wash step was repeated another time. The beads were resuspended in 500 µl Lysis Buffer 2 followed by addition of 5 µg of the appropriate antibody and were incubated for 2 hr on a nutator at 4°C. The beads were collected with a magnetic stand and the supernatant was aspirated followed by two wash steps in 500 µl Lysis Buffer 2. The nuclear extracts were added to the beads and incubated with the beads on a nutator at 4°C overnight. After the incubation the beads were collected with a magnetic stand and the supernatant aspirated. 1 ml Lysis Buffer 2 was added and the sample was mixed gently by inverting the tube until all beads were fully resuspended. The beads were then collected with a magnetic stand and the supernatant aspirated. This wash step was repeated another two times (three washes total). The beads were resuspended

in 50 µl of 1 x SDS Laemmli Buffer and the inputs with 15 µl 4 x SDS Laemmli Buffer and all samples were boiled for 5 min at 95°C on a heating block (Thermo Fisher Scientific, 88870003). Inputs were centrifuged in a table top centrifuge for 5 min at full speed while bead IP samples were centrifuged for 2 min at 850 g at RT. Equal volumes of Input and IP samples were loaded and separated on a 4–20% gradient SDS-PAGE gel (Bio-Rad, 4561096 and 4561093) in running buffer (Bio-Rad, 1610772) using a Bio-Rad electrophoresis and blotting system (Bio-Rad, 1658033). Western blotting was carried out as described under 'Western blotting'.

Lysis Buffer 1:
  10 mM HEPES pH 7.9; 10 mM KCl; 1.5 mM $MgCl_2$; 0.5% Igepal CA-630 (NP-40); 0.5 mM DTT; protease inhibitors (Sigma, P8340) (1:200).
Lysis Buffer 2:
  20 mM HEPES pH 7.9; 420 mM NaCl; 1.5 mM $MgCl_2$; 0.2 mM EDTA; 10% glycerol; 0.5 mM DTT; protease inhibitors (Sigma, P8340) (1:200).
4 x SDS Laemmli Buffer:
  250 mM Tris pH 6.8, 50% Glycerol, 8% SDS, 0.008% Bromophenol blue.
1 x SDS Laemmli Buffer:
  900 µl 4 X SDS Laemmli Buffer Stock + 100 µl β-Mercaptoethanol.

## Proliferation assay

$5 \times 10^4$ mESCs (WT, *Dnttip1* KO and *Elmsan1* KO) per well were plated in a 24-well plate and proliferation was monitored over a period of 5 days. From day 2 onward mESCs were harvested and counted with an automated cell counter (Thermo Fisher Scientific, Countess II FL Automated Cell Counter). For each time point and clone experiments were repeated in triplicate. For statistical analysis unpaired Student's t-test was performed between WT and *Dnttip1* KO/*Elmsan1* KO mESCs.

## qPCR

qPCR reactions were carried out with SYBR Green PCR Master Mix (Thermo Fisher Scientific, 4309155) on a QuantStudio 7 Flex Real-Time PCR System (Thermo Fisher Scientific) in 384-well format (Thermo Fisher Scientific, 4309849) in a total reaction volume of 10 µl with 1 µl of undiluted eluted ChIP DNA and a final primer concentration of 200 nM. qPCR primers used in this study are shown in *Supplementary file 6*. The calculation and analysis for % Input was as follows *Dahl and Collas (2008)*: % Input=[(Amount of ChIP DNA)/(Amount of Input DNA x Dilution Factor)] x 100.

## RNA isolation

RNA was isolated from $3 \times 10^6$ cells with the RNeasy Mini Kit (Qiagen, 74106) following the manufacturer's instructions with the following alterations. Cells were resuspended in 600 µl RLT buffer (with 2-Mercaptoethanol) and the homogenate further passed through a QiaShredder column (Qiagen, 79656) by centrifugation in a table top centrifuge for 2 min at full speed at RT. The optional step after the second wash with RPE buffer was applied to dry the membrane. RNA was eluted with 85 µl $H_2O$ and supplied with 10 µl 10 x DNAase buffer and 5 µl DNAse I (NEB, M0303S), mixed and incubated at RT for 20 min. After incubation RNA was purified by following the 'RNA Cleanup' protocol in the RNeasy Mini Handbook. The optional step after the second wash with RPE buffer was applied to dry the membrane. RNA was eluted in 50 µl $H_2O$ and concentration determined with a NanoDrop 8000 Spectrophotometer.

## RNA-seq library preparation and sequencing

RNA was quantified using the Quant-iT RiboGreen RNA Assay Kit (Thermo Fisher Scientific, R11490) and quality-checked with the RNA 6000 Nano Kit (Agilent, 5067–1511) on a 2100 Bioanalyzer (Agilent, G2939BA) or High Sensitivity RNA ScreenTape Assay (Agilent, 5067–5579, 5067–5580, 5067–5581) on a 4200 TapeStation (Agilent, G2991AA) prior to library generation. Libraries were prepared from total RNA with the TruSeq Stranded Total RNA Library Prep Gold Kit (Illumina, 20020599) according to the manufacturer's instructions. Libraries were analyzed for insert size distribution with the High Sensitivity DNA Kit (Agilent, 5067–4626) on a 2100 Bioanalyzer or the High Sensitivity D1000 ScreenTape Assay (Agilent, 5067–5584, 5067–5585, 5067–5587, 5067–5603) on a 4200 TapeStation. Libraries were quantified using the Quant-iT PicoGreen dsDNA Assay (Thermo Fisher

Scientific, P11496). Paired end 100 cycle sequencing was performed on a HiSeq 2500, HiSeq 4000, or NovaSeq 6000 System (all from Illumina) according to the manufacturer's instructions.

## qRT-PCR

qRT-PCR reactions were carried out with the *Power* SYBR Green RNA-to-C$_T$ *1-Step* Kit (Thermo Fisher Scientific, 4389986) on a QuantStudio 7 Flex Real-Time PCR System (Thermo Fisher Scientific) in 384-well format (Thermo Fisher Scientific, 4309849) in a total reaction volume of 10 µl with 40 ng of RNA and a final primer concentration of 200 nM. qRT-PCR primers used in this study are shown in *Supplementary file 5*. Analysis was performed applying the delta-delta Ct method ($2^{-\Delta\Delta Ct}$) as previously described (*Livak and Schmittgen, 2001*).

## Western blotting

Whole cell lysates from mESCs or differentiated neuroectoderm were obtained according to the following protocol. Medium was aspirated, DPBS was added and cells were detached by resuspension with a pipet or a cell scraper and transferred into a fresh tube. After centrifugation the supernatant was aspirated and the pellet resuspended in 1 ml ice cold DPBS. After repelleting in a cooled table top centrifuge and aspiration of the supernatant the cell pellet was resuspended in RIPA buffer followed by incubation on a nutator for 30 min at 4°C. The resulting cell lysate was centrifuged in a cooled table top centrifuge for 5 min at full speed and transferred to a new tube. The protein concentration of the supernatant was determined by a protein assay (BioRad, 5000006) with a spectrophotometer (Eppendorf, 6133000010) using disposable cuvettes (Eppendorf, 0030079353). The protein concentration was adjusted to a desired final protein concentration of 1–2 µg/µl with 4 x SDS Laemmli Buffer and additional RIPA Buffer to achieve a final 1 x SDS Laemmli Buffer concentration. 1 x SDS Laemmli Buffer samples were boiled for 5 min at 95°C on a heating block (Thermo Fisher Scientific, 88870003) and centrifuged in a table top centrifuge for 5 min at full speed at RT. From the supernatant equal amounts of protein (usually 10–30 µg) were loaded and separated on a 4–20% gradient SDS-PAGE gel (Bio-Rad, 4561096 and 4561093) in running buffer (Bio-Rad, 1610772) using a Bio-Rad electrophoresis and blotting system (Bio-Rad, 1658033). Proteins were transferred to either a PVDF or nitrocellulose membrane (Santa Cruz Biotechnology, sc-3723 and sc-3724) with the same Bio-Rad system using Western Transfer Buffer (Bio-Rad, 1610771 and 20% methanol) for 90 min at 400 mA. The membranes were blocked for 1 hr at RT with gentle rocking in 5% dry milk in TBST buffer. Primary antibody incubation was performed overnight at 4°C with gentle rocking in 5% dry milk in TBST Buffer. On the next day the membranes were washed three times for 5 min with 10 ml TBST Buffer with gentle rocking and then incubated with horse radish peroxidase coupled secondary IgG-specific antibodies in TBST Buffer for 1 hr at room temperature with gentle rocking. After an additional three washes for 5 min with 10 ml TBST Buffer with gentle rocking the membranes were developed using Immobilon Crescendo Western HRP Substrate (Millipore, WBLUR0500) and imaged on an Odyssey Fc imaging system (LI-COR, Model: 2800). Buffer compositions and antibodies that were used and their concentrations are listed below. For histone western blots cells were trypsinized, resuspended in growth medium and pelleted by centrifugation. The pellet was resuspended in DPBS and cells were counted with an automated cell counter (Thermo Fisher Scientific, Countess II FL Automated Cell Counter). Equal numbers of cells were pelleted, resuspended in 1 x SDS Laemmli Buffer, boiled for 10 min at 95°C on a heating block and centrifuged in a table top centrifuge for 5 min at full speed at RT. From the supernatant equal volumes (usually 5–15 µl) were loaded and separated on a 4–20% gradient SDS-PAGE gel. The remainder of the workflow was identical to the one for the RIPA buffer extracted samples described above.

Antibody dilutions for western blot applications:

Mouse α-Actin (Developmental Studies Hybridoma Bank [DSHB], JLA20); 1:1000 (supernatant). Mouse α-DNTTIP1 (Novus Biologicals, NBP2-02507); 1:500. Rabbit α-DNTTIP1 (Bethyl Laboratories, A304-048A); 1:2000. Rabbit α-DAB1 (Cell Signaling Technology, 3328); 1:5000. Rabbit α-pDAB1 (Cell Signaling Technology, 3327); 1:1000. Rabbit α-ELMSAN1 (This study, 34421); 1:5000. α-FAK (Thermo Fisher Scientific, 39–6500); 1:1000. α-pFAK (Thermo Fisher Scientific, 700255); Rabbit α-H3 (Abcam, ab1791); 1:50000. Rabbit α-H3K4ac (RevMAb, 31-1063-00); 1:1000. Rabbit α-H3K27ac (RevMAb, 31-1056-00); 1:1000. Rabbit α-H3K79ac (RevMAb, 31-1052-00); 1:1000. Rabbit α-H4 (Abcam, ab10158); 1:50000. Rabbit α-H4K20ac (RevMAb, 31-1084-00); 1:1000. Rabbit α-HDAC1 (Cell Signaling Technology, 34589); 1:2000.

Rabbit α-HDAC2 (Cell Signaling Technology, 57156); 1:2000. Rabbit α-MASH1 (Abcam, ab74065); 1:5000. Sheep α-NETRIN1 (R&D Systems, AF6419); 1:1000. Mouse α-PAX6 (DSHB, AB_528427); 1:500. Goat α-ROBO3 (R&D Systems, AF3155); 1:500. Rabbit α-SLIT3 (Abcam, ab186706); 1:1000. Mouse α-UNC5B (R&D Systems, MAB1006); 1:1000.

Western blotting buffers:
RIPA Buffer:
25 mM Tris pH 7.5; 150 mM NaCl; 1 mM EDTA; 1% Triton X-100; 0.1% SDS; 0.1% Na-deoxy-cholate; protease inhibitors (Sigma, P8340) (1:100).
SDS-PAGE Running Buffer:
25 mM Tris pH 8.3, 192 mM glycine, 0.1% SDS.
Western Transfer Buffer:
25 mM Tris pH 8.3, 192 mM glycine, 20% methanol.
TBST Buffer:
20 mM Tris pH 7.5, 150 mM NaCl, 0.1% Tween 20.
4 x SDS Laemmli Buffer Stock:
250 mM Tris pH 6.8, 50% Glycerol, 8% SDS, 0.008% Bromophenol blue.
1 x SDS Laemmli Buffer:
900 µl 4 X SDS Laemmli Buffer Stock + 100 µl β-Mercaptoethanol.

## CRISPR/Cas9 genome editing

Genetically modified C57/BL6 Bruce-4 mESCs were generated using CRISPR/Cas9 technology. Briefly, 400000 C57/BL6 Bruce-4 mESCs grown in feeder free culture conditions (KnockOut DMEM medium) were transiently co-transfected with 500 ng of gRNA expression plasmid (Addgene, 43860), 1 µg Cas9 expression plasmid (Addgene, 43945), and 200 ng of pMaxGFP via nucleofection (Lonza, 4D-Nucleofector X-unit) using solution P3 and program CA137 in small (20 µl) cuvettes according to the manufacturer's recommended protocol. Cells were single cell sorted by FACS to enrich for GFP-positive (transfected) cells, clonally selected and verified for the desired targeted modification via targeted deep sequencing. Three clones were identified for each modification, assessed in relevant assays and then transitioned to chemically defined naïve culture conditions (2iL). Two clones for each gene knock-out were further used for more specific assays in this study. The sequences for each gRNA and relevant primers are listed in *Supplementary file 1*.

## Quantification and statistical analysis

### ChIP-seq analysis

#### Mapping reads and visualizing data

ChIP-seq raw reads were aligned to the mouse and *Drosophila melanogaster* hybrid reference genomes (mm9+dm3) using BWA (version 0.7.12; default parameters) and duplicated reads were then marked with Picard (version 1.65), with only nonduplicated reads kept by samtools (version 1.3.1, parameter ''-q 1 -F 1024''). Mapped reads were then split into two bam files (mapped to mm9 and dm3 respectively). For data quality control and to estimate the fragment size, the nonduplicated version of SPP (version 1.11) was used to calculate the relative strand correlation value with support of R (version 3.3.1). To visualize ChIP-seq data on the integrated genome viewer (IGV) (version 2.3.82), we utilized genomeCoverageBed (bedtools 2.25.0) to obtain genome-wide coverage in BEDGRAPH file format and then converted it to bigwig file format by bedGraphToBigWig. The big-wig files were scaled to 15 million reads to allow comparison across samples.

#### Peak calling, annotation and motif analysis

MACS2 (version 2.1.1 20160309) was used to call narrow peaks (DNTTIP1, HDAC1, H3K27ac and H3K4me3) with option 'nomodel' and 'extsize' defined as fragment size estimated by SPP and a FDR corrected p-value cutoff of 0.05.SICER (version 1.1, with parameters of redundancy threshold 1, window size 200 bp, effective genome fraction 0.86, gap size 600 bp, FDR 0.00001 with fragment size defined above) was used for broad peak/domain calling (H3K4me1 and H3K27me3). Enriched regions were identified by comparing the ChIP library file to input library file. Peak regions were defined to be the union of peak intervals from two ChIP replicates of WT, *Dnttip1* KO or *Elmsan1* KO mESCs, respectively. Promoter regions were defined as ±1000 bp from a TSS based on the

mouse RefSeq annotation. Genomic feature annotation of peaks was carried out by annotatePeaks. pl, a program from HOMER suite (v4.8.3, http://homer.salk.edu/homer/). HOMER software was used to perform de novo motif discovery and to check for enrichment of known motifs from a set of DNTTIP1 peaks (associated with up- or downregulated genes in *Dnttip1* KO versus WT mESCs) or all DNTTIP1 peaks.

### Spike-in normalization and differential analysis

ChIP-seq raw read counts were reported for each region/each sample using bedtools 2.25.0. Spike-in normalization was performed by counting *Drosophila* reads and mouse reads in each ChIP sample and corresponding input sample and using those counts to generate a normalization factor for each sample, which was calculated as (ChIP_dm3.reads/ChIP_mm9.reads)/(Input_dm3.reads/Input_mm9. reads). Raw read counts were voom normalized and statistically contrasted using the pipeline limma in R (version 3.3.1). The normalization factor defined above was used to modify the mouse library size in edgeR (version 3.16.5) for CPM calculation and differential analysis. An empirical Bayes fit was applied to contrast *Dnttip1* KO and *Elmsan1* KO samples to WT samples and to generate $\log_2$ fold changes, p-values and false discovery rates for each peak region. Histograms showing average ChIP-seq intensity over gene bodies were generated using ngsplot (v2.61).

### RNA-seq analysis

Total stranded RNA sequencing data were generated and mapped against mouse genome assembly NCBIM37.67 using the StrongArm pipeline described previously (*Wu et al., 2016*). Gene level quantification values were obtained with HT-seq based on the GENCODE annotation (vM20) and normalized by the TMM method with edgeR (version 3.16.5). Differential expression analysis was performed with the voom method applying the limma pipeline in R (version 3.3.1). Significantly up- and down- regulated genes were defined by at least a 1.5 fold change in gene expression and a p-value<0.01. Reactome and gene set enrichment analysis (GSEA) were carried out using GSEA or EnrichR, respectively. Gene expression $\log_2$ CPM (counts per million) values were computed for heatmap and box plot visualization. $\log_2$ FPKM gene expression values were applied for bar plot diagrams.

### Neurite and neuron analysis

The neuron-specific markers TUBB3 and MAP2 were used to identify neurons by IF as described under 'Neural differentiation'. NeuriteTracer, a plugin of ImageJ (version 1.52a), was used to manually trace the length and number of neurites per neuron and to automatically detect nuclei based on DAPI staining using the IF raw image data for each genotype. The length of neurites was determined from 200 neurons for each genotype. The number of neurites per neuron was determined manually for a total of 200 neurons per genotype. The percentage of neurons within the total cell population in NE was calculated manually by determining the number of MAP2-positive neurons in relation to the total number of cells as determined by DAPI staining or alternatively by flow cytometry analysis of TUBB3/DAPI-stained NE cell populations. Neurite length was classified into shorter neurites (<50 μm) and longer neurites (≥50 μm) and the mean length was compared amongst relevant conditions by applying the Student's t-test. % of neurons = (number of neurons per image as assessed by MAP2 staining/total number of cells per image as assessed by DAPI staining) x 100. For each analysis, experiments were carried out in triplicate. An unpaired Student's t-test as well as ONE-way ANOVA was performed for each of the treated conditions and compared either to the WT or to their respective controls.

### Statistical analysis of qPCR and qRT-PCR data

qPCR signals for manual ChIPs were calculated as % of input (% Input) from technical duplicates. Error bars represent the standard deviation from technical duplicates. For qRT-PCR relative gene expression levels were calculated applying the delta-delta Ct method ($2^{-\Delta\Delta Ct}$). Error bars depict the standard deviation from technical triplicates. Statistical significance was determined by applying the Student's unpaired t-test.

## Public datasets and additional bioinformatics analysis

HDAC1 ChIP-seq data from WT mESCs was obtained from gene expression omnibus (GEO): GSE55437. Data was analyzed as described under 'Quantification and statistical analysis' within the 'ChIP-seq analysis' section if applicable.

## Acknowledgements

We thank BaoHan Vo and Martine Roussel for generously providing granule neuron progenitor cells; Alyssa House, Scott Perry, Mohona Sarkar, and Richard Ashmun from the Flow Cytometry and Cell Sorting Shared Resource for conducting cytometry and FACS analyses; Dana Roeber, Rain Sun, Sanchit Trivedi, Scott Olsen, and Geoffrey Neale from the Hartwell Center for RNA- and ChIP-seq library preparation, sequencing, and support; and Jamshid Temirov for advice and guidance with microscopy-related applications. We thank all Herz lab members for insightful comments and critical reading of the manuscript. This work was supported by a transition to independence grant from the National Institutes of Health/National Cancer Institute (R00CA181506) to H-MH and the American Lebanese Syrian Associated Charities (ALSAC).

## Additional information

### Funding

| Funder | Grant reference number | Author |
| --- | --- | --- |
| National Cancer Institute | R00CA181506 | Hans-Martin Herz |
| American Lebanese Syrian Associated Charities | | Hans-Martin Herz |

The funders had no role in study design, data collection and interpretation, or the decision to submit the work for publication.

### Author contributions

Baisakhi Mondal, Conceptualization, Formal analysis, Validation, Investigation, Visualization, Methodology, Writing - original draft, Writing - review and editing; Hongjian Jin, Data curation, Software, Formal analysis, Validation, Investigation, Visualization, Methodology; Satish Kallappagoudar, Yurii Sedkov, Tanner Martinez, Formal analysis, Validation, Investigation, Visualization; Monica F Sentmanat, Resources, Formal analysis, Validation, Investigation, Visualization; Greg J Poet, Supervision, Methodology, Writing - review and editing; Chunliang Li, Data curation, Software, Formal analysis, Supervision, Investigation, Visualization, Methodology; Yiping Fan, Resources, Data curation, Software, Formal analysis, Supervision, Validation, Investigation, Visualization, Methodology; Shondra M Pruett-Miller, Hans-Martin Herz, Conceptualization, Resources, Formal analysis, Supervision, Funding acquisition, Validation, Investigation, Visualization, Methodology, Project administration, Writing - review and editing

### Author ORCIDs

Baisakhi Mondal (ID) https://orcid.org/0000-0002-8171-4333
Greg J Poet (ID) https://orcid.org/0000-0002-9923-5218
Chunliang Li (ID) http://orcid.org/0000-0002-5938-5510
Shondra M Pruett-Miller (ID) http://orcid.org/0000-0002-3793-585X
Hans-Martin Herz (ID) https://orcid.org/0000-0003-4780-9176

### Decision letter and Author response

Decision letter https://doi.org/10.7554/eLife.57519.sa1
Author response https://doi.org/10.7554/eLife.57519.sa2

# Additional files

## Supplementary files

• Supplementary file 1. List of guide RNA and deep sequencing primer sequences used for CRISPR/Cas9-mediated genome editing of *Dnttip1* and *Elmsan1* in mESCs.

• Supplementary file 2. Differentially expressed genes in *Dnttip1* KO (KO1 and KO2) and *Elmsan1* KO (KO1 and KO2) mESCs (FC $\leq -1.5$ or $\geq 1.5$; p-value<0.01) (MS Excel spreadsheets).

• Supplementary file 3. DNTTIP1 binding sites identified by ChIP-seq in mESCs (MS Excel spreadsheet).

• Supplementary file 4. Genes that are up- or downregulated in *Dnttip1* KO (KO1 and KO2) versus WT mESCs (FC $\leq -1.5$ or $\geq 1.5$; p-value<0.01 except for Spry4) and bound by DNTTIP1 (MS Excel spreadsheet).

• Supplementary file 5. List of primer sequences used for qRT-PCR analysis.

• Supplementary file 6. List of primer sequences used for ChIP qPCR analysis.

• Transparent reporting form

## Data availability

RNA-sequencing and ChIP-sequencing data have been deposited in GEO under the accession code GSE131062. All data generated or analyzed during this study are included in the manuscript and supporting files.

The following dataset was generated:

| Author(s) | Year | Dataset title | Dataset URL | Database and Identifier |
|---|---|---|---|---|
| Mondal B, Jin H, Kallappagoudar S, Sedkov Y, Martinez T, Sentmanat MF, Poet GJ, Li C, Fan Y, Pruett-Miller SM, Herz H-M | 2020 | The histone deacetylase complex MiDAC regulates a neurodevelopmental gene expression to control neurite outgrowth | https://www.ncbi.nlm.nih.gov/geo/query/acc.cgi?acc=GSE131062 | NCBI Gene Expression Omnibus, GSE131062 |

The following previously published dataset was used:

| Author(s) | Year | Dataset title | Dataset URL | Database and Identifier |
|---|---|---|---|---|
| Lee BK, Shen W, Lee J, Rhee C, Chung H, Kim KY, Park IH, Kim J | 2015 | TG-interacting factor1 (Tgif1) maintains the identity of mouse ES cells by counterbalancing the expression of core pluripotency factors and ES cell core factors | https://www.ncbi.nlm.nih.gov/geo/query/acc.cgi?acc=GSE55437 | NCBI Gene Expression Omnibus, GSE55437 |

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
