## [Decision Letter]

[Editors' note: this paper was reviewed by Review Commons.]

**Acceptance summary:**

This high-quality analysis presents the first physiological role for the MiDAC HDAC complex. You clearly show the mechanistic basis of this function at the level of gene expression and histone modifications. This will make an important impact on the field.

---

## [Author Response]

We thank the reviewers for their time to review the manuscript and their helpful comments. Below in a point by point response we have fully addressed all of the points raised by the reviewers and have also visibly implemented these changes in a revised version of our manuscript accordingly.

Reviewer #1 (Evidence, reproducibility and clarity):The authors show MiDAC is required for the neurodevelopmental programme spanning from neuronal outrgowth to guidance mechanisms. They compare WT, Dnttip1 KO and Elmsan1 KO and they perform a series of RNAseq and CHIPseq studies followed by functional and biochemical assays in differentiating neurons.The evidence is strong and the data are well presented and overall convincing1) It would be great to see how this mechanism responds to neurotrophin, cross communicates with related epigenetic regulators and if modification of guidance affects neuronal behavior in vivo or in in vitro assays such as stripe assays for example.

We fully agree with the reviewer’s suggestion that future studies should include in vitro and in vivo experiments that further analyze additional aspects of MiDAC function in neurogenesis and should also address how MiDAC relates to other epigenetic regulators in this context. As this is the first publication reporting a physiological function for MiDAC, there is lots of potential for follow-up studies. We are particularly interested in MiDAC’s cross-communication with other epigenetic regulators and are actively working on this aspect.

2) It would be good to make clear the biological replicates use for each RNAseq and CHIPseq experiments in the figure legends.

We have updated all relevant figure legends accordingly to indicate which biological replicates from our RNA-seq and ChIP-seq experiments are depicted in the individual figure panels.

Reviewer #1 (Significance):The manuscript provides an important advance to both the neuroscience and epigenetics community. I am a neurobiologist working in epiegenetics.Reviewer #2 (Evidence, reproducibility and clarity):Mondal and colleagues present a comprehensive analysis of the biological function of the poorly understood HDAC complex MiDAC. In contrast to the other HDAC1/2 containing complexes (NuRD, SIN3 and CoREST), MiDAC's function is poorly understood. The authors show that in mESCs Midac comprises DNTTIP1, ELMSAN1, TRERF1, ZNF541 and HDAC1 (but not HDAC2). Generation of mESC KO lines revealed that DNTTIP1 and ELMSAN1 are key for the integrity and targeting of MiDAC. Comparing the mESCs and their derived KO lines, the authors show that MiDAC binds promoters and enhancers genome wide and regulates a neurodevelopmental gene expression program. Key targets are identified and functional studies show their importance in neurite outgrowth. Surprisingly, MiDAC has a dualistic function, both activating (through H4K20 deacetylation) and repressing (through H3K27 deacetylation).This manuscript describes for the first time a role for MiDAC in regulating neurodevelopment, while finding no evidence for a function in cell cycle control. The comprehensive experiments and data are of excellent quality, convincing and fully support the conclusions of the authors.1) I have no demands for extra experiments. However, the authors might want to consider the role of HDAC1. It appears that MiDAC is unique in only harboring HDAC1, rather than both HDAC1 and HDAC2 (as SIN3, NuRD and CoREST). Thus it might be possible to exploit this redundancy in the other HDAc complexes to identify MiDAC-specific functions and targets for HDAC1.

We are grateful for the reviewer’s insightful comment. Indeed, in mouse ES cells a preference exists for HDAC1 to be incorporated into MiDAC (our data). However, other groups have previously shown that in other cell types both HDAC1 and HDAC2 can be incorporated into MiDAC (Bantscheff et al., 2011; Hao et al., 2011). We concur with the reviewer that at least in a cell type-specific context such as mouse ES cells this will open up an interesting avenue to identify MiDAC-specific targets and functions for HDAC1.

2) The proposed direct role for H4K20 deacetylation in transcriptional activation, while interesting, remains somewhat speculative. The Discussion and Abstract should reflect this

We thank the reviewer for pointing this out and have now more carefully phrased our proposed role for H4K20 deacetylation in transcriptional activation in the Abstract and Discussion, respectively.

3) The authors discuss the potential role of MiDAC in human neurodevelopmental disorders. Did they check available databases for MiDAc mutations associated with such disorders?

While we were not able to find any obvious association of mutations in MiDAC components in autism databases individual noncoding mutations in several MiDAC components have recently been reported in autism patients (Zhou et al., 2019). We now reference this article in the Discussion accordingly.

Minor comments:4) The MIDAC composition in Figure 7 needs to be made more clear. The letting is also very small.

We have updated Figure 7 and the figure legend accordingly to better highlight the composition of MiDAC and increased the letter size for easier readability.

5) The impressive amount of data in this MS at some places reads like a shopping list. Imposing more hierarchy, recaps and conclusions might improve structure and readability.

We appreciate the reviewer’s suggestion and have modified the manuscript at select locations to accommodate readability especially as it pertains to recaps and conclusions.

Reviewer #2 (Significance):The data are compelling and the findings are of general interest. This work reveals a mechanistic insight into a novel connection between an epigenetic regulator and (neuro)development. This work also generates valuable hypothesis and questions for follow-up research.